# Drone honey bees are disproportionately sensitive to abiotic stressors despite expressing high levels of stress response proteins

Alison McAfee [1,2,3 ✉], Bradley N. Metz [1,2 ✉], Joseph P. Milone[1], Leonard J. Foster [3] & David R. Tarpy[1,2]

Drone honey bees (*Apis mellifera*) are the obligate sexual partners of queens, and the availability of healthy, high-quality drones directly affects a queen's fertility and productivity. Yet, our understanding of how stressors affect adult drone fertility, survival, and physiology is presently limited. Here, we investigated sex biases in susceptibility to abiotic stressors (cold stress, topical imidacloprid exposure, and topical exposure to a realistic cocktail of pesticides). We found that drones (haploid males) were more sensitive to cold and imidacloprid exposure than workers (sterile, diploid females), but the cocktail was not toxic at the concentrations tested. We corroborated this lack of cocktail toxicity with in-hive exposures via pollen feeding. We then used quantitative proteomics to investigate protein expression profiles in the hemolymph of topically exposed workers and drones, and found that 34 proteins were differentially expressed in exposed drones relative to controls, but none were differentially expressed in exposed workers. Contrary to our hypothesis, we show that drones express surprisingly high baseline levels of putative stress response proteins relative to workers. This suggests that drones' stress tolerance systems are fundamentally rewired relative to workers, and susceptibility to stress depends on more than simply gene dose or allelic diversity.

[1] Department of Entomology & Plant Pathology, North Carolina State University, Raleigh, NC 27695, USA. [2] Department of Applied Ecology (current), North Carolina State University, Raleigh, NC 27695-7617, USA. [3] Department of Biochemistry and Molecular Biology, Michael Smith Laboratories, University of British Columbia, Vancouver, BC V6T1Z4, Canada. ✉email: amcafee@ncsu.edu; bnmetz@ncsu.edu

High quality male honey bees (*Apis mellifera* drones) are essential for supporting adequate mating of queens, whose longevity depends on the number and quality of sperm acquired during nuptial flights[1]. Despite being critical players in honey bee reproduction, factors affecting drone quality are generally understudied (reviewed recently by Rangel et al.[2]), as the vast majority of existing studies have focused on workers (reviewed in Chmiel et al.[3]), the sterile female caste.

Existing research has shown that adult drone exposure of some pesticides[4–9] and extreme temperatures[10–13] negatively impact drone fertility. Other research has focused on drone pesticide exposure during larval development[14], but generally, little is known about how adult drones' tolerance to abiotic stressors and their stress-mitigating responses compare to workers. Kairo et al.[8] identified a negative effect of fipronil on drone fertility but found no affect on survival; however, exact exposure levels were unknown because drones were exposed indirectly through foraging workers. Grassl et al.[15]. found that drones are more sensitive to thiamethoxam than workers, and our previous research shows that drones are more susceptible than workers to heat[10].

This apparent biased sensitivity of haploid male bees to abiotic stressors may be in part explained by an extension of the haploid susceptibility hypothesis, which states that haploid individuals are more susceptible to pathogenic infections, since they have no opportunity for allelic diversity that comes with heterozygosity[16]. O'Donnell and Beshers, who proposed the haploid susceptibility hypothesis[16], described the notion as a corollary to the well-known heterozygous advantage[17]. While existing examples of haploid susceptibility are described in the context of disease and parasites, in theory, the notion could equally apply to abiotic stressors as well.

The haploid susceptibility hypothesis is not consistently supported when it comes to pathogenic infections, and has been challenged[18]. While investigations on honey bee male susceptibility to *Nosema*[19], as well as immunocompetence of leafcutter ants (*Atta colombica*)[20], wood ants (*Formica exsecta*)[21], and buff-tailed bumble bees (*Bombus terrestris*)[22] support the haploid susceptibility hypothesis, research on *B. terrestris* male susceptibility to *Crithidia* does not[18]. However, this hypothesis has generally not been discussed in the context of abiotic stress, despite also being a relevant challenge.

Neonicotinoid pesticides have been described as inadvertent insect contraceptives, owing to evidence that thiamethoxam and clothianidin can reduce drone fertility and lifespan during colony-level oral exposures to low concentrations of insecticides (<5.0 parts-per-billion (ppb))[5]. Furthermore, topical exposure of queens to 2 μl of 20 ppb imidacloprid, another neonicotinoid, has been shown to reduce viability of sperm stored within the queens[23]. Friedli et al. found that thiamethoxam and clothianidin had a greater impact on developmental stability of drones compared to workers, and attribute this pattern to be driven in part by male haploidy[4].

Experiments documenting effects of exposure to specific classes of pesticides are important; however, since drones do not forage, they are most likely to encounter more complex pesticide mixtures that accumulate in hive matrices (e.g., wax, pollen, honey). Furthermore, because drones encounter contaminated wax, consume pollen and honey, and experience temperature variation in the hive and on mating flights, we expect that they, like workers, should experience some selection for abiotic stress tolerance mechanisms. They also experience indirect selection through their sister workers (on average, 25% of a drone's genes are present in his sisters) and mother queen (100% of a drone's genes are present in his mother).

In a survey of commercial honey bee colonies in the U.S., Traynor et al. documented residue data for pesticides, herbicides, and fungicides present in beebread, wax, and other hive components[24,25]. These data offer a realistic reference point for investigating effects of compound cocktails in relevant abundances and proportions. While high residue concentrations within hive matrices have been linked to queen failure[24,26], which was likely driven by indirect effects on worker jelly secretions rather than direct effects on the queen[27,28], the impact of this realistic cocktail on drone physiology has not yet been investigated.

Here, we aim to investigate drone and worker tolerances to abiotic stressors, focussing mainly on pesticide exposure. Our overarching goal was to investigate the response of putative stress response proteins that could potentially underly sex biases in tolerance. We confirm drone susceptibility to imidacloprid, and we further investigate the impacts of drone exposure to pesticide cocktails (based on data in Traynor et al.[24]) through topical applications as well as supplemental hive treatments. Finally, we investigate drone and worker stress responses to topical pesticide applications (control, imidacloprid, and cocktail treatments) through proteomics analysis of the hemolymph. Given that drones are generally considered more sensitive to pesticide exposure than workers, which our data corroborates, we used these data to address our predictions that (1) drones have lower constitutive expression of relevant stress response proteins, and (2) that workers, but not drones, elevate detoxification enzymes in response to exposure. Our data suggest that drones have surprisingly strong baseline expression of putative stress response proteins, contrary to our expectations, causing us to re-evaluate exactly why drones appear to be so intolerant to abiotic stress. Regardless of the mechanism, we suggest that, in light of male sensitivity to abiotic stressors in honey bees, as documented here, the haploid susceptibility hypothesis may be expanded to include abiotic stress in addition to pathogens and parasites. Experiments investigating sex biases in other social insects are needed to determine how generalizable these observations are.

## Results

**Sex biases in survival across abiotic stressors.** We and others have previously reported drone-biased sensitivity to heat[10,11], and Grassl et al. identified drone-biased sensitivity to thiamethoxam[15]. To determine if this sex bias exists across other abiotic stressors, we used cage experiments to compare worker and drone sensitivity to cold (4 °C, which acts as a positive control for mortality), imidacloprid (1, 10, and 100 ppm, 2 μl topical exposure in acetone), and a cocktail of the nine compounds frequently found in wax (as described in McAfee et al.[29], recipe derived from Traynor et al.[24], administered at 0.33x, 2x, and 10x, where x is the median concentration in wax). Bees were sourced from three different colonies, but we found no effect of colony source on survival so this parameter was dropped from the statistical model. Baseline drone survival in the negative control groups ranged from 91 to 93% in all three experiments for the duration of the test (survival was measured after two days), whereas worker survival was 100%. As expected, we found a strong sex bias in cold tolerance; no workers perished in the experiment, but drone survival counts were significantly affected by cold exposure ($z = -3.6$, df = 45, $p = 0.00031$; generalized linear model, binomial distribution), with 76 and 92% of drones perishing after 2 and 4 h exposures at 4 °C, respectively (Fig. 1a). Worker survival was also not affected by imidacloprid exposure at the tested doses and the number of replicates ($z = -0.01$, df = 115, $p = 0.99$). Drone survival, however, was significantly affected by imidacloprid dose ($z = -1.99$, df = 115, $p = 0.047$; Fig. 1b). These are highly unrealistic exposure scenarios and are strictly employed to investigate sex-biases. No appreciable drone

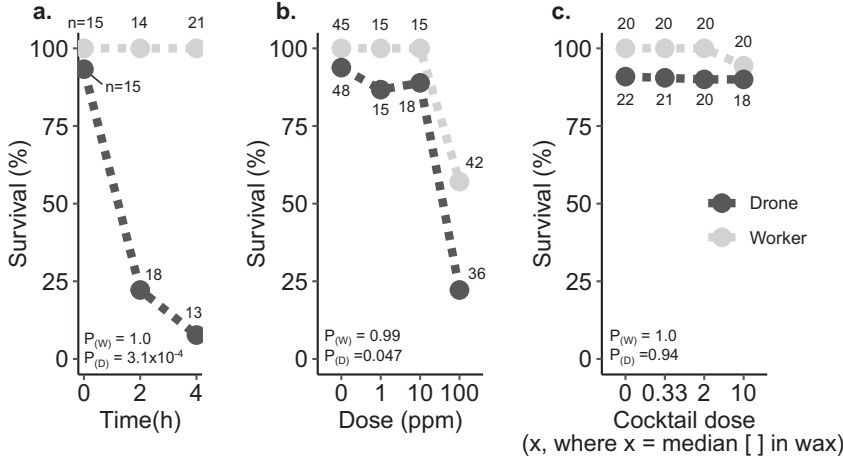

**Fig. 1 Sex biases in survival during temperature and pesticide stress challenges.** Drones and workers (all five days old at the beginning of the experiment) from three different colony sources were paint marked and kept in California-style queen shipping cages with candy (one subject with five other companion workers in each cage). Statistical differences were evaluated using a generalized linear model on survival count data with a binomial distribution. Sample sizes for each group are indicated in the figure. Baseline survival rates are indicated by the negative control group (2 µl topical acetone treatment for 'cocktail' and 'imidacloprid' groups, and room temperature incubation for the 'cold' group) after two days. **a** Survival of drones and workers two days after exposure to different durations of cold stress (4 °C), **b** topical exposure to different concentrations of imidacloprid, and **c** topical exposure to different concentrations of a pesticide cocktail of compounds commonly found in wax (see McAfee et al.[29] for the recipe, and Traynor et al.[24] for the supporting data). See Supplementary Data 1 for underlying data.

or worker mortality was observed with exposure to any cocktail dose (workers: $z = -0.003$, df = 76, $p = 1.0$; drones: $z = -0.079$, df = 81, $p = 0.94$), indicating that the agrichemical matrix commonly found in wax has low contact toxicity to both male and female bees (Fig. 1c).

**Equivocal effects of in-hive cocktail exposures on drone survival, body size, and fertility.** Since the agrichemical cocktail is the exposure that drones are most likely to directly experience in a managed setting, and prior evidence suggests that exposure through pollen poses a greater hazard than wax[28], we aimed to corroborate our negative results of topical cocktail exposure with hive exposures via pollen patties, targeting either drone adults (experiment 1) or drone larvae (experiment 2). In the first experiment, we banked naïve adult drones either in colonies that had been previously fed a pesticide pollen patty supplement for 28 d, or in colonies that were fed pollen patties with no added pesticides. We found that drones banked in control colonies actually had greater mortality than colonies fed patties containing the pesticide cocktail ($\chi^2_1 = 19.8$; $p < 0.0001$). However, this trend was only significant for drones from one of the two source colonies (source one: $\chi^2_1 = 0.0846$; $p = 0.771$; source two: $\chi^2_1 = 38.2$; $p < 0.0001$) and this is further confounded by significant differences among mortality within the banks within each treatment or source group (minimum $\chi^2_2 = 11.7$, $p = 0.003$; Fig. 2a).

We next investigated effects of colony source, bank treatment, and their interaction on drone fertility (see "Methods" for a definition of this parameter). We found that drone size differed only across colony source (linear mixed model, $t_{237.1} = 3.72$; $p = 0.0002$), with neither an effect of treatment ($t_{14.4} = 2.09$; $p = 0.055$), nor source by treatment interaction ($t_{237} = -1.56$; $p = 0.12$), or bank colony (likelihood ratio = 1.23; $p = 0.267$; Fig. 2b). Testing the same model for drone fertility revealed a significant treatment by source interaction ($t_{237} = 2.99$; $p = 0.003$), so we then tested each source separately. Fertility of drones from source one did not differ because of treatment ($t_6 = -0.197$; $p = 0.85$) or adult bank (likelihood ratio = 3.63; $p = 0.057$). However, fertility of drones from source two differed due to treatment ($t_6 = 2.90$; $p = 0.027$) but not adult bank

(likelihood ratio = 1.18; $p = 0.28$), with the trend of drones banked in treated colonies actually having higher fertility (Fig. 2c).

We expected that worker care at the larval stage could have a greater or more consistent effect on drones, as has been observed for queens previously[27,30]. Therefore, in experiment two, we investigated the effects of colony cocktail exposure via pollen patties on the quality of drones they reared. We first fed colonies either pesticide or control pollen patties for 28 d, then inserted empty drone frames to be laid out and continued to feed the patties throughout subsequent drone development. We found that drone emergence numbers were not significantly different across treatment ($F_{1,25} = 3.08$; $p = 0.09$) though colonies ranged widely in drones produced (0–120 individuals). Drone mortality followed a similar trend as in experiment 1 with a strong trend toward higher mortality among the control treatment ($\chi^2_1 = 47.0$; $p < 0.0001$) driven by strong differences in mortality among control sources ($\chi^2_2 = 84.3$; $p < 0.0001$) rather than treatment sources ($\chi^2_1 = 1.3 \times 10^{-30}$; $p = 1$) and a clear confounded factor of near total mortality of a single control source (Fig. 2d). Drones differed in size based solely on the colony source (likelihood ratio = 68.9; $p < 0.0001$) rather than larval treatment ($t_5 = -0.09$; $p = 0.93$; Fig. 2e). The results were similar for drone fertility, with significant differences due to source (likelihood ratio = 9.2; $p = 0.002$), but not larval treatment ($t_5 = 0.04$; $p = 0.97$; Fig. 2f).

**Sex-biased protein expression patterns.** Unlike responses of the cocktail treatments, sex-biased tolerance to imidacloprid was apparent. To investigate the molecular origin of workers' and drones' responses to pesticides, we performed differential protein expression analysis on hemolymph from workers and drones exposed to the control (acetone, $N = 6$ workers and $N = 8$ drones), cocktail ($10x$, $N = 8$ drones and $N = 8$ workers), or imidacloprid (1 ppm, $N = 7$ drones and $N = 7$ workers) treatments. This is not meant to investigate the effects of realistic exposures (a 1 ppm topical imidacloprid exposure is very high); rather, the purpose is to determine what proteins are involved in sex-biased stress tolerance. We did not examine cold-stressed drones because too few drones survived the treatment to analyze.

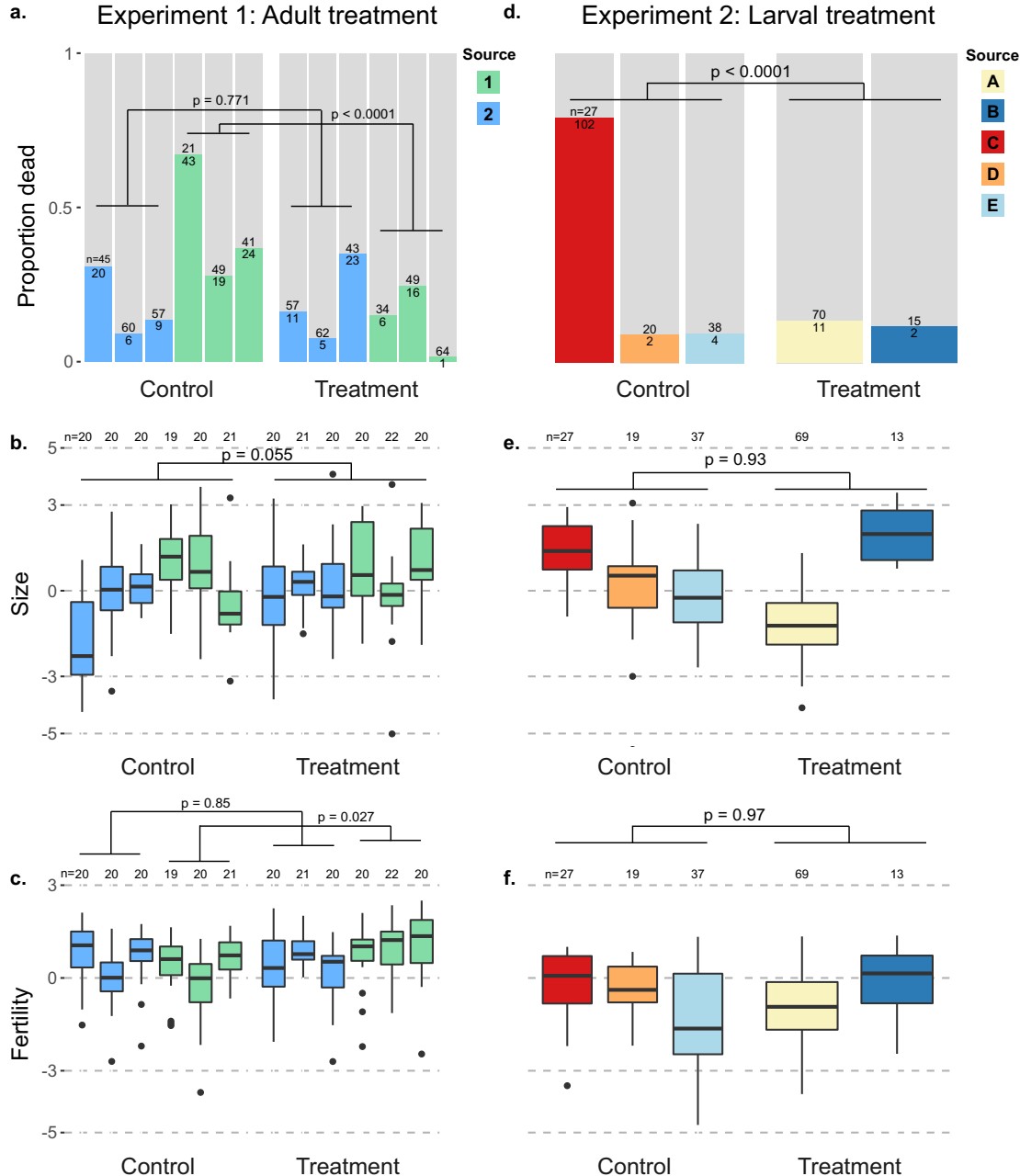

**Fig. 2 No consistent effects of hive-level cocktail treatments via pollen on drone development or adult fostering.** Within colonies, drones were exposed to pesticide-laced pollen patties either as adults (experiment 1) or as larvae (experiment 2). Unique drone source colonies are differentiated by colour. Mortality data are reported as proportion dead presented in colour, with live proportion presented in grey. Mortality differences were evaluated with $\chi^2$ tests, whereas size and fertility were evaluated using linear mixed models (see Methods for specific models). Boxes represent the interquartile range, bars indicate the median, and whiskers span 1.5 times the interquartile range. Sample sizes (*n*) are printed at the column break for the mortality tests (**a** and **d**) and along the top of each box plot (**b**, **c** and **e**, **f**). **a**–**c** Mortality, size, and fertility of drones from different source colonies reared in untreated colonies but fostered in either treated or untreated colonies as adults. **d**–**f** Mortality, size, and fertility of drones from different source colonies which were reared through development by treated and untreated colonies, but fostered in untreated colonies as adults. See Supplementary Data 2 for underlying morphometric data and Supplementary Data 3 for survival data.

We identified 1452 protein groups in total (1% false discovery rate (FDR)), but after filtering out proteins without at least three identifications in each experimental group, 654 proteins groups were quantified. Of those, 188 were differentially expressed between drones and workers, and 34 were differentially expressed between control drones and imidacloprid-treated drones (Fig. 3a, b, all at 5% FDR, Benjamini–Hochberg correction). No differences were identified in any of the other pairwise comparisons, including cocktail treatments relative to controls, as well as all

exposed worker comparisons, further supporting that the cocktail is not hazardous to drones at these doses and under these conditions. Of the 34 proteins differentially expressed between imidacloprid- and acetone-treated drones, 18 were also differentially expressed between drones and workers (Fig. 3c–e).

We hypothesized that drones might be disproportionately sensitive to abiotic stressors if they express low levels of stress-mitigating proteins (e.g., detoxification enzymes). We therefore expected that putative stress response proteins would be

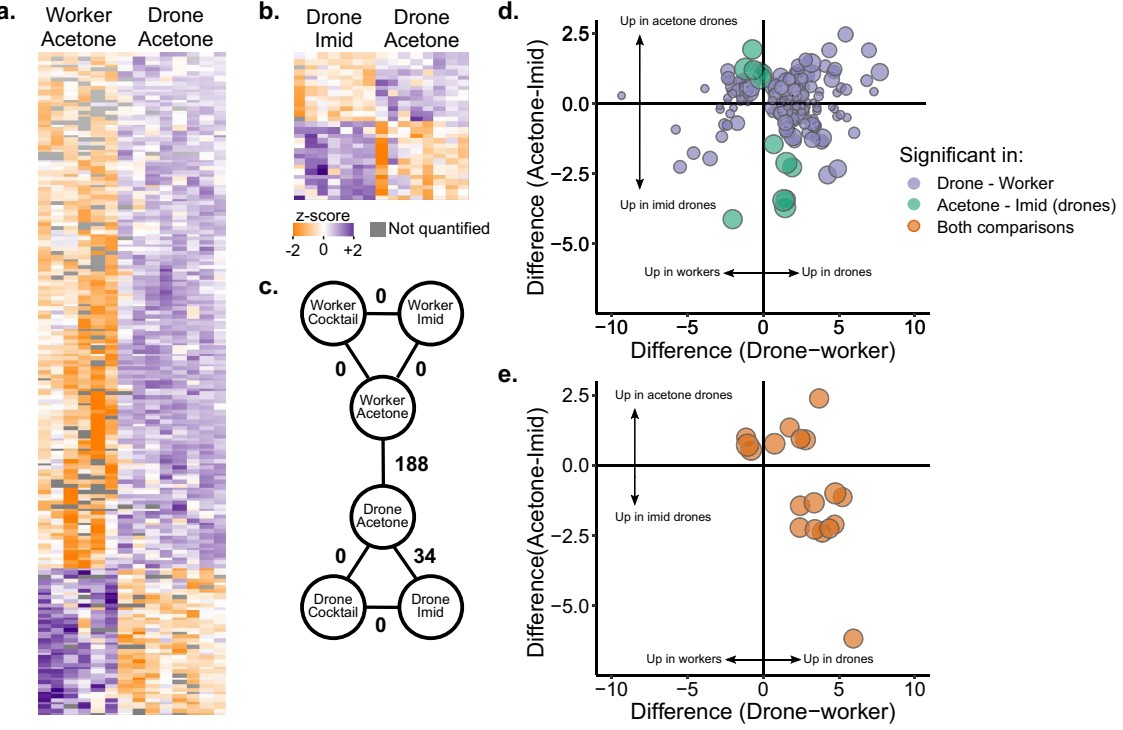

**Fig. 3 Quantitative proteomics on surviving drone and worker hemolymph after topical exposure to pesticides.** Workers and drones exposed to the pesticide cocktail (2 µl of 10x solution in acetone, where x = the median concentration in wax), imidacloprid (2 µl of 1 ppm solution in acetone), and a negative control (2 µl of acetone). **a** Label-free quantitative proteomics results comparing negative control drones and workers. Each row is a protein and each column is a sample. Only significant differences (5% FDR, Benjamini Hochberg correction) are depicted. **b** Proteins differentially expressed between imidacloprid-exposed drones and controls. **c** Summary of statistical analyses comparing different groups. Numbers indicate the number of proteins differentially expressed at 5% FDR. **d** Summary of significance and direction of change in expression for proteins differentially expressed in drones vs. workers (purple), and imidacloprid vs. control drones (green). Circle size is proportional to −log(p value) for the drone imidacloprid vs. acetone comparison. **e** Proteins differentially expressed in both statistical comparisons. Circle size is proportional to −log(p value) for the drone imidacloprid vs. acetone comparison. Underlying data can be found in Supplementary Data 4, and summary statistics of these data are in Supplementary Data 5.

expressed at lower levels in drones relative to workers. Interestingly, we observed the opposite trend. All but two of 17 putative stress response proteins, including proteins linked to detoxification, oxidative stress, immunity, DNA repair, and heat-shock proteins, were upregulated in control drones relative to control workers (Fig. 4a; 5% FDR, Benjamini–Hochberg correction). Among these were three heat-shock proteins (HSP cognate 3, HSP beta 1, and 97 kDa HSP, corresponding to NP_001153524.1, XP_003251576.1, and XP_006561225.1, respectively), which were all expressed more highly in drones than workers, but were not differentially regulated by pesticide exposure (Fig. 4b). An uncharacterized protein (XP_026295805.1) with high sequence homology to *Drosophila melanogaster* glutathione-S-transferase S1, a detox-ification enzyme, was also upregulated in drones relative to workers, and downregulated in imidacloprid-treated drones relative to the negative control (Fig. 4c). Furthermore, glutathione-S-transferase S4 (XP_006560566.1) was one of the top three upregulated proteins in drones relative to workers, with a log2(fold change) of 6.97. HSP70 Ab (NP_001153544.1) was differentially expressed in both sex and pesticide-exposure comparisons, with higher expression in control drones over control workers, and still higher expression in imidacloprid-exposed drones over control drones. Further-more, expression of the small heat-shock protein pl(2)el (XP_006568238.2) increased with imidacloprid treatment for drones, but not workers (Fig. 4d). HSP60 (XP_392899.2) was one of the two putative stress response proteins that were downregulated in drones relative to workers, and it and its

binding partner, HSP10 (XP_624910.1), were both further downregulated with imidacloprid treatment (Fig. 4e).

Drones appear to exhibit a robust suite of stress response proteins even in the absence of temperature or pesticide stress, and we hypothesized that a dramatic mobilization of resources would be required to sustain these basal expression levels. Hexamerins are well-known amino acid storage and juvenile hormone-binding proteins that are highly abundant in larval hemolymph and are thought to be catabolized to support metamorphosis, when the developing bee is unable to feed[31–33]. We expected that adult drones would have low levels of hexamerins in their hemolymph, in order to support the energetically costly maintenance of high basal levels of proteins involved in stress responses. There are four major honey bee hexamerins: hex70a, hex70b, hex70c, and hex110[32]. Only hex70a (NP_001104234.1) and hex110 (NP_001094493.1) were quantified in our dataset. Whereas hex70a did not exhibit sex biased expression patterns, hex110 was actually the most strongly differentially expressed protein between drones and workers, with significantly lower abundance in drones (p = 0.000244, q = 0.00296, t = −5.32; Fig. 5a), which supports our hypothesis. After hex110, the top 4 proteins exhibiting the strongest sex-biased expression were serpin88Ea (XP_026298978.1), trypsin 1-like (XP_026301257.1), inositol-3-phosphate synthase (XP_623377.1), and adenylate kinase (NP_001164443.1; Fig. 5b).

**Accounting for the potential effect of acetone treatment.** Because our experimental design did not include untreated con-trols, it is possible that drones appear to have high baseline levels

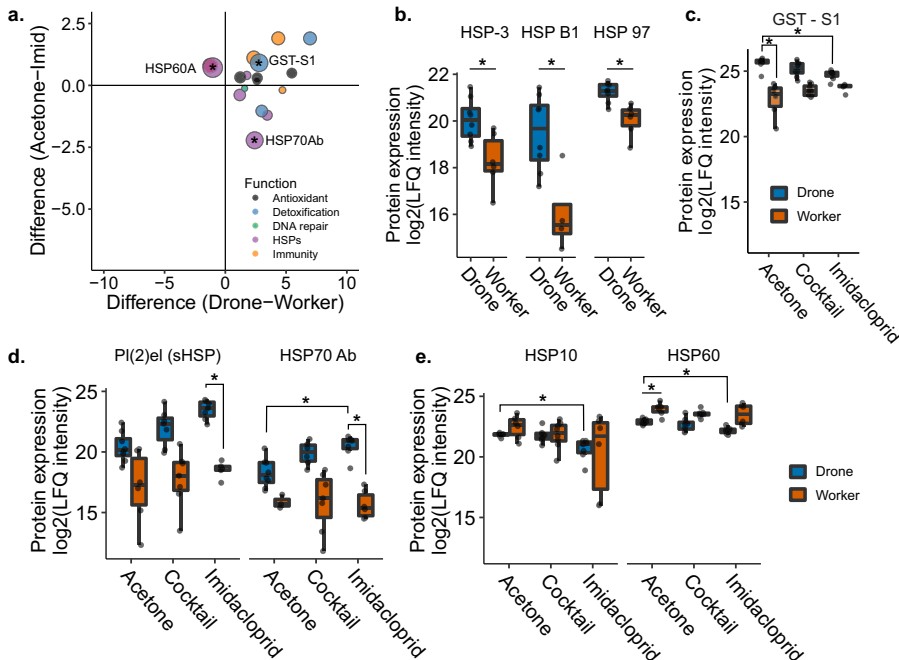

**Fig. 4 Differentially expressed proteins linked to stress responses to pesticides. a** Proteins differentially expressed between drones and workers which have functions linked to oxidative stress, detoxification, DNA repair, the heat-shock response, and immunity. Asterisks indicate that the protein was also differentially expressed in drone imidacloprid vs. control comparisons. Boxes represent the interquartile range, bars indicate the median, and whiskers span 1.5 times the interquartile range. Circle size is proportional to $-\log(p$ value) for the drone imidacloprid vs. acetone comparison. **b** Patterns of expression for HSP cognate 3, HSP beta 1, and 97 kDa HSP in drones and workers. Asterisks indicate that the comparison was significant at 5% global FDR (Benjamini Hochberg). **c** Pattern of expression of an uncharacterized protein best matching glutathione-S-transferase S1 in *Drosophila melanogaster* across pesticide treatments. Asterisks indicate that the comparison was significant at 5% global FDR (Benjamini Hochberg). **d** Patterns of expression of protein lethal (2) essential for life (Pl(2)el) and HSP70 Ab across pesticide treatments. Asterisks indicate that the comparison was significant at 5% global FDR (Benjamini-Hochberg). **e** Patterns of expression of 10 kDa HSP and HSP60, mitochondrial HSPs which are known to physically interact in a 1:1 stochiometric ratio. Asterisks indicate that the comparison was significant at 5% global FDR (Benjamini–Hochberg). Underlying data can be found in Supplementary Data 4, and summary statistics of these data are in Supplementary Data 5.

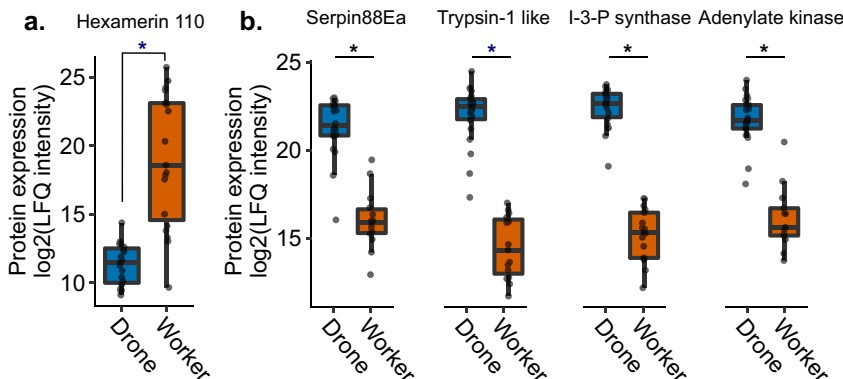

**Fig. 5 Expression of the top five most strongly differentially expressed proteins, in terms of fold-change, between drones and workers.** Expression of these proteins was not linked to pesticide exposure so the exposure data are included in the drone and worker categories. Asterisks indicate that the comparison was significant at 5% global FDR (Benjamini–Hochberg). Boxes represent the interquartile range, bars indicate the median, and whiskers span 1.5 times the interquartile range. **a** Hex110, a conserved amino acid storage protein, was the most strongly differentially expressed protein overall, with very low levels present in the drones. **b** The top four other proteins were all more abundant in the drones, including a serine protease inhibitor (serpin88Ea), a serine protease (trypsin-1 like), inositol-3-phosphate synthase, and adenylate kinase. Underlying data can be found in Supplementary Data 4, and summary statistics of these data are in Supplementary Data 5.

of stress response proteins relative to workers simply because they have a stronger response to acetone (the negative control). To determine if this could be the case, we compared hemolymph proteomes from age-matched, untreated drones and workers ($n = 14$ each) sampled from three different hives. Of the 483 proteins that were differentially expressed (out of 988 proteins quantified after filtering), we found that the data largely agree with the findings above. Namely, glutathione-S-transferases were again significantly upregulated in drones (Fig. 6a), hex110 was again strongly downregulated in drones whereas inositol-3-phosphate synthase was upregulated (Fig. 6b), and HSPs were largely also upregulated in drones with the exception of HSP

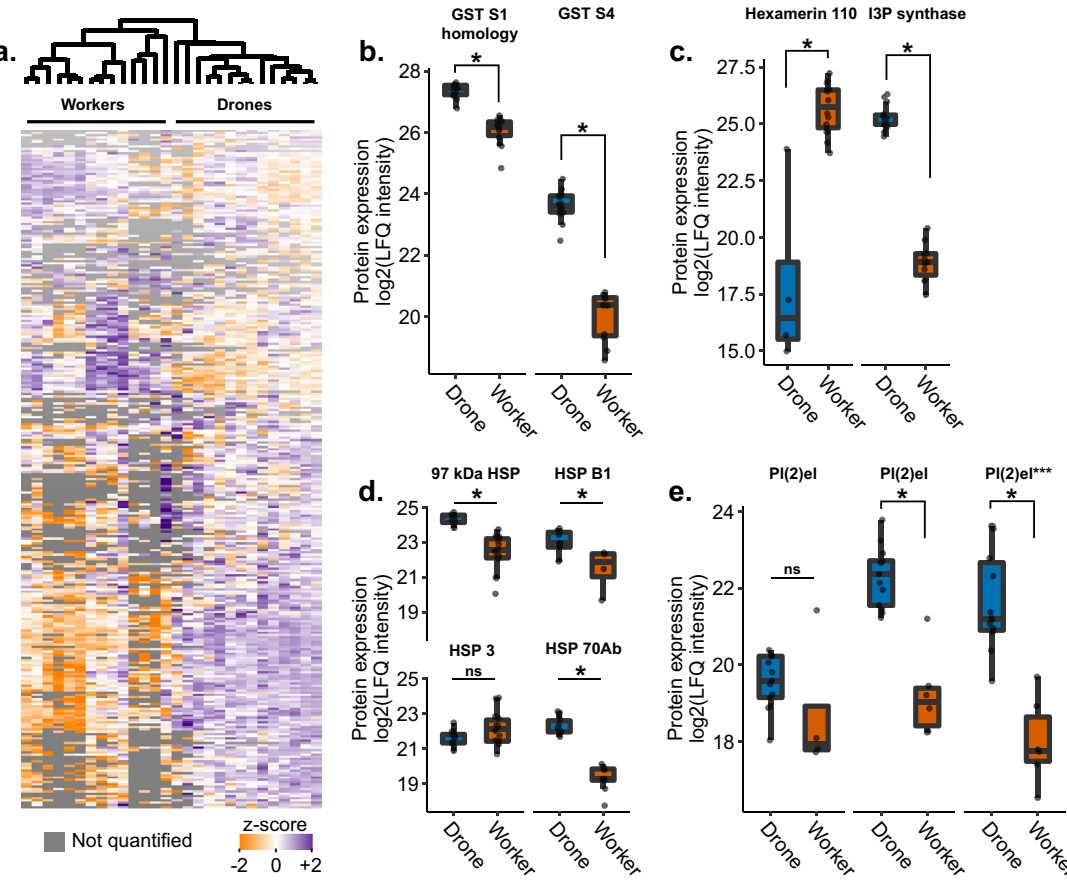

**Fig. 6 Comparing untreated drones and workers.** Boxes represent the interquartile range, bars indicate the median, and whiskers span 1.5 times the interquartile range. **a** Quantitative proteomics identified 2,089 proteins in the hemolymph of $n = 14$ drones and $n = 14$ workers, with 988 quantified after quality filtering. **b–e** 483 proteins were differentially expressed (two group comparison, Benjamini–Hochberg correction at 5% FDR), with proteins of interest highlighted here. **e** Several proteins share the name protein lethal(2) essential for life-like (pl(2)el), and the specific pl(2)el, which was depicted in Fig. 4d is highlighted by three asterisks here. Underlying proteomics data are available in Supplementary Data 6.

cognate 3 and one of the three proteins named pl(2)el (Fig. 6c, d). These results are consistent with our initial findings and suggest that the data are not artifacts of acetone treatment or caging.

## Discussion

The haploid susceptibility hypothesis states that haploid individuals are more susceptible to parasites and pathogens because their haploid state reduces allelic diversity while also providing no opportunity for heterozygous compensation should an unfavourable allele be possesed[4,16]. While we did not examine the presence of mutations in these bees, we did show that adult haploid honey bees (drones) are indeed more susceptible to cold stress and pesticide stress than diploid worker bees, similar to heat stress and thiamethoxam exposure as has been demonstrated previously[10,11,15]. We, therefore, suggest that the haploid susceptibility hypothesis is likely broadly applicable to abiotic as well as biotic stressors, as originally proposed[16].

We further investigated the impact of drone exposure to a realistic agrochemical cocktail by feeding pesticide-laced pollen patties to whole colonies, but found no consistent effects of pesticide treatment on adult drones or larvae. The data indicate that this mixture either does not have appreciable toxicity under the tested conditions, or that drones are buffered from exposure by indirect feeding through workers. Finally, we identified a surprising general trend for drones to express higher levels of putative stress response proteins compared to workers—findings

consistent with the trends observed in the existing honey bee protein atlas comparing sexes, castes, and tissues[34,35]—and some of the same proteins were also differentially regulated by drones in response to pesticide stress.

This strong expression of stress response proteins was in contrast to hex110 expression, which is a major amino acid storage protein[32] that was massively downregulated in drones relative to workers. We tentatively propose that mobilization of hex110 may provide the resources needed for drones to express significantly higher constitutive levels of stress response machinery compared to workers, and that this scenario may represent a long-term sacrifice (depletion of hex110 stores) for the sake of a short-term gain (high constitutive expression of stress response proteins). This hypothesis will require further testing, such as with hex110 knock-down experiments, in order to concretely ascertain.

Our results suggest that the haploid susceptibility hypothesis does not fully explain the general sensitivity of drones to stress, since, from a protein abundance standpoint, the drones' repertoire of proteins that help mitigate damage due to stressors is surprisingly robust. Nor does gene dose, per se, explain the trends observed, since drones have half the gene dose as workers yet express higher basal levels of stress response proteins. Rather, we suggest that the drone's stress response may be a result of sex-specific rewiring of the stress responses, such that most of their amino acid reserves are constantly mobilized to support high baseline levels of proteins that help mitigate damage due to short-

term stress. Indeed, drones have concentrations of free amino acids in the hemolymph that are over three times higher than in workers of the same age[36]. This suggests that, rather than being stored as hexamerins, these resources are mobilized, perhaps to support other sex-specific protein expression. One of the top five differentially expressed proteins was an adenylate kinase, an enzyme that plays a central role in cellular energy homeostasis, cell proliferation, and AMP-induced cell signalling; thus, it is a prime candidate to investigate as a master regulator of sex-specific metabolic rewiring.

It is somewhat confusing for drones to so easily die in the face of challenging conditions if their stress response is already primed. One explanation is that, while they generally express high levels of stress response proteins, they have no further amino acid or energy reserves to amplify that response, should a severe intensity or duration of stress be encountered. We suggest that the drone's investment in high baseline expression proteins linked to oxidative damage, detoxification, temperature stress, DNA damage, and immune protein expression enables them to combat a wide range of mild stressors, but leaves them unable to launch a stronger response to deal with intense, specific stressors.

Conversely, these results also raise the question of how workers are so stress-tolerant, in terms of survival, without launching an equally robust stress response. Indeed, we identified no differentially expressed proteins comparing workers treated with imidacloprid to controls. While investigating worker responses to an entomopathogenic fungus, Bull et al. suggest that immune inducibility is not a reliable indicator of resistance, and it is possible that the same may be true with regard to abiotic stress responses. However, Bull et al. also propose that higher baseline expression of immune proteins in foragers relative to workers may explain why foragers are more resistant, despite exhibiting low inducibility. This creates a conundrum, because workers are generally more stress-tolerant, and we identified both low inducibility and low baseline expression of stress response proteins relative to drones. Enzyme abundance also does not necessarily correlate with detoxification activity in honey bee workers[30,37]. On the whole, these data indicate that differential expression is difficult to interpret in the absence of protein activity, but the question remains: Why do drones express such high levels of putative stress response proteins, if they ostensibly remain inactive?

Since we only quantified 654 protein groups and many proteins exist below our limit of detection, it is possible that important stress response proteins were simply not quantified in our dataset. However, we still expected to see at least some sign of a stress response in workers. An alternate explanation is that, since these bees were euthanized two full days after experiencing the stress, it is possible that workers are more efficient in their stress response than drones, and have already both launched and reversed their stress response. This may be done through differential expression, or in a manner mediated by post-translational modification of proteins to modulate the proteins' specific activities. These explanations would be consistent with the ability for workers to rapidly and efficiently mobilize, then shut down, a specific stress response as conditions change.

In the data presented here, we show that numerous heat-shock proteins were differentially expressed in drones relative to workers, as well as imidacloprid-exposed drones relative to unexposed drones. While drones expressed higher levels of HSP beta 1, HSP cognate 3, 97 kDa HSP, protein lethal(2) essential for life, and HSP70 Ab, and a putative glutathione-S-transferase, they expressed lower levels of HSP60 compared to workers. Drones also tended to express lower levels of HSP10 (the binding partner of HSP60)[38]. HSP60 and HSP10 assist with folding proteins imported to the mitochondria and prevent protein aggregation[38,39]. HSP10 and HSP60 are further

downregulated with imidacloprid treatment, indicating that imidacloprid exposure may make drones even more susceptible to heat stress than they already are. Although not reflected in our data, HSP70 Ab and HSP cognate 3 have previously been found to become downregulated in honey bee workers in response to imidacloprid exposure[40]. The overlap between proteins differentially expressed with heat[10,29,41] and imidacloprid exposure suggest that these environmental stressors may have interactive effects. Furthermore, HSPs have been demonstrated to have antiviral effects in workers[42], yet heat-shock appears to repress other humoral immune genes[43], suggesting further interactions between heat-stress and immunity. Determining the additive and synergistic effects of multi-stressor drone exposure as well as indirect effects of drone exposure on queens (such as work done by Kairo et al.[8,44] and Bruckner et al.[45]) is an important study area to broaden.

Among the proteins most strongly differentially regulated between drones and workers were serpin 88Ea, trypsin-1 like, inositol-3-phosphate (I-3-P) synthase, and adenylate kinase (Fig. 5). This is puzzling, as serpin88Ea is a serine protease inhibitor and trypsin-1 like is a serine protease, therefore, their strong co-expression appears to be an inefficient use of resources. However, whether trypsin-1 like is actually a target of serpin88Ea is unknown. In *Drosophila*, serpin88Ea is a negative regulator of the toll immune response via inhibition of spaetzle processing enzyme[46], and in honey bees, serpin88Ea has been linked to stored sperm viability[47], therefore, it is possible that these two proteins are actually involved in different processes. I-3-P synthase, however, is an enzyme known to become upregulated with abiotic stress in the closely related Eastern honey bee (*Apis cerana*), and which in turn regulates antioxidant enzymes such as superoxide dismutase and glutathione-S-transferase[48]. Ni et al.[48] found that knockdown of *A. cerana* I-3-P synthase subsequently inhibited antioxidant enzyme expression, for example. Given that our data show that drones consistently upregulate other stress response proteins, is likely that the enzyme also has a similar function in Western honey bees too. Finally, adenylate kinase plays an important role in regulating cellular energy homeostasis and various signalling pathways[49]. Its strong upregulation in drones may further point to drones operating on the margins of energy expenditure.

The in-hive drone exposures to a pesticide cocktail were meant to further investigate the null result we obtained when topically exposing drones to different concentrations of a pesticide blend. We reasoned that, similar to previous work conducted on queens and royal jelly production[27,28], an oral pollen exposure may affect drones where a direct topical exposure does not, potentially via altered worker care or jelly secretions. However, we did not observe consistent effects of pesticide treatment on drone size or fertility, whether the colonies were exposed during the drones' development or adulthood. Rather, we observed that the drone source colony had the most pronounced effect on these quality parameters, indicating that, at the very least, any colony level effects of cocktail exposure that might exist are far outweighed by other natural colony parameters. We note that different results may be obtained for bees from a different genetic stock or which are experiencing combined effects of other stressors.

Overall, we found that drones are more sensitive, in terms of survival, to cold stress and imidacloprid exposure compared to workers, and that drones exhibit surprisingly strong constitutive expression of putative stress response proteins. These proteins have a variety of functions, including detoxification, DNA repair, immunity, and oxidative stress. With some exceptions, these proteins are generally expressed more strongly in drones relative to workers, which, coupled with very low levels of hex110 (a major amino acid storage protein) in drones, suggests that drones favour broad, non-specific upregulation of putative

**Table 1 Sample sizes and colony origins for drone and worker survival comparisons.**

| Stressor | Sex | Colony | # Bees |
|---|---|---|---|
| Imidacloprid | Worker | Farm 002 | 26 |
| | Drone | Farm 002 | 30 |
| | Worker | Roof 002 | 20 |
| | Drone | Roof 002 | 13 |
| | Worker | Roof 004 | 19 |
| | Drone | Roof 004 | 13 |
| | Worker | SL NE | 25 |
| | Drone | SL NE | 32 |
| | Worker | SL SE | 27 |
| | Drone | SL SE | 29 |
| Cocktail | Worker | Farm 002 | 26 |
| | Drone | Farm 002 | 31 |
| | Worker | SL NE | 22 |
| | Drone | SL NE | 26 |
| | Worker | SL SE | 30 |
| | Drone | SL SE | 26 |
| Cold | Worker | Farm 002 | 17 |
| | Drone | Farm 002 | 18 |
| | Worker | SL NE | 15 |
| | Drone | SL NE | 16 |
| | Worker | SL SE | 18 |
| | Drone | SL SE | 13 |

stress response proteins to the detriment of retaining amino acid reserves. We speculate that this may improve their likelihood of protection against mild stressors but impair their ability to withstand prolonged or intense stress, since they have few reserved resources to draw on. Future research should focus on effects of multi-stressor exposures on drones and genetic variability of stress tolerance in the population, which are major gaps in knowledge of honey bee reproduction.

## Methods

**Drone and worker survival**. In May, newly emerged (callow) drones and workers from three different colonies located in Vancouver, Canada, were marked with paint pens (Posca, Japan) and allowed to age for five days in their respective colonies. On day five, they were collected and placed in wooden California mini queen cages (approximately 7 cm × 2.5 cm × 1.3 cm with one open face covered by a mesh screen) containing fondant in a candy tube. Each painted drone was placed in a separate cage with five other young (non-flying) worker bees, which were not analyzed in the experiment, to attend the drones. Workers were housed in the same way. To select attendants, the bees were picked up and dropped ~ 30 cm back onto the frame to determine their propensity to fly. Different stress tests were conducted on different days, and the number of aged bees available differed for each sampling, but in all cases, there was roughly equivalent representation from each colony. See Table 1 for sample sizes originating from different colony sources and Supplementary Data 1 for a complete description. Colonies were headed by genetically unrelated queens which were produced the summer prior to experimentation and had successfully overwintered. The colonies "Farm 002," "SL NE," and "SL SE" had two standard deep hive bodies whereas colonies "Roof 002" and "Roof 004" were single box colonies. All colonies were treated for *Varroa* mites using Apivar the previous fall, but spring mite treatments were delayed until after drones and workers were collected. Only bees free of *Varroa* at emergence were marked.

To analyze pesticide stress, bees were briefly anesthetized with carbon dioxide to immobilize them, then 2 μl of pesticide active ingredient solution (either a cocktail mixture or imidacloprid, which was not part of the cocktail, in acetone) was applied directly to the thorax as an acute exposure. The pesticide cocktail mixture (which contained tau-fluvalinate, coumaphos, 2,4-DMPF (2,4-Dimethylphenyl-N′-methyl-formamidine, a degradation product of amitraz), chlorothalonil, chlorpyriphos, fenpropathrin, atrazine, pendimethalin, and azoxystrobin) was produced exactly as previously described[29] based on wax residue data published by Traynor et al.[24]. We tested doses of 0x, 0.33x, 2x, and 10x, where x is the median concentration in wax found in commercial colonies in the U.S. (see Table 2 for components and their concentrations). As previously described, all compounds in the pesticide mixture were purchased as pure technical material (≥95% purity) from Sigma Aldrich (St. Louis, MO) or Chem Service Inc. and were serially diluted in acetone in order to achieve the respective concentrations[29]. The imidacloprid solutions were produced by serial dilution of the technical chemical acquired from Chem Service Inc. (West

Chester, PA). We tested doses of 0, 1, 10, and 100 ppm. In addition to the pesticide challenge, we also tested acute cold stress susceptibility by placing the caged bees in a covered container in a 4 °C refrigerator for 0, 2, or 4 h.

After all treatments, bees were allowed to recover for two days at room temperature in the dark, and were provided with two drops of water (~ 100 μl) per day. After the two day stress recovery period, we counted the number of bees that were alive and dead. Workers and drones from the 10x cocktail group and 1 ppm imidacloprid group were euthanized by submerging in ethanol and then frozen at −20 °C for one week, until protein extraction. Cold stressed drones were not analyzed because survival was prohibitively poor.

**Proteomics sample preparation**. Hemolymph for the pesticide stress experiment was extracted from frozen bees by allowing them to thaw on ice, then using a scalpel to make a small incision between their 2nd and 3rd abdominal tergites (see Table 3 for sample sizes). We inserted a small glass capillary against the incision to collect the hemolymph, then expelled the hemolymph (about 1–2 μl for workers, about 5–8 μl for drones) into 50 μl of 100 mM Tris buffer (pH 8.0). This solution was mixed and then spun at $16,000 \times g$ to remove cellular debris, then transferred to a new tube. Clarified solution was precipitated using acetone as previously described[47,50] (final acetone concentration = 80%, incubated at −20 °C overnight). Precipitated protein was pelleted by spinning at $10,000 \times g$ for 15 min, the supernatant was discarded, and the pellet washed with 500 μl of 80% ice cold acetone. The supernatant was discarded and the pellet was dried at room temperature prior to suspending in digestion buffer (6 M urea, 2 M thiourea, 100 mM Tris, pH 8). Protein concentration was determined with a Bradford assay, and samples were prepared for mass spectrometry exactly as previously described[29]. Briefly, the samples were reduced, alkylated, digested with Lys-C, diluted with five volumes of 50 mM ammonium bicarbonate, then digested with trypsin overnight. Peptides were desalted using C18 STAGE tips[51] as previously described, quantified via nanodrop, and 1 μg of peptides were analyzed in randomized order on an Easy nLC-1000 (Thermo) chromatography system connected to a Bruker Impact II Q-TOF mass spectrometer.

As stated in McAfee et al.[10,29], the LC system included a fused-silica (5 μm Aqua C18 particles (Phenomenex)) fritted 2 cm trap column connected to a 50 cm analytical column packed with ReproSil C18 (3 μm C18 particles (Dr. Maisch)). The separation gradient ran from 5 to 35% Buffer B (80% acetonitrile, 0.1% formic acid) over 90 min, followed by a 15 min wash at 95% Buffer B (flow rate: 250 μl/min). The instrument parameters were: scan from 150 to 2200 m/z, 100 μs transient time, 10 μs prepulse storage, 7 eV collision energy, 1500 Vpp Collision RF, a + 2 default charge state, 18 Hz spectral acquisition rate, 3.0 s cycle time and the intensity threshold was set to 250 counts.

Mass spectrometry data were searched as previously described[29,52] using MaxQuant (v 1.6.1.0) with default parameters, except that match between runs and label-free quantification were enabled. The data were searched against the honey bee reference proteome available on NCBI (based on the build HAv3.1, downloaded Nov 18th, 2019) with all honey bee virus and *Nosema* proteins included in the fasta file, which is available with the raw data hosted on MassIVE (www.massive.ucsd.edu; MSV000087818)[53]. The proteinGroups.txt file was imported to Perseus (v 1.6.1.1), where proteins only identified by site, potential contaminants, and reverse hits were removed, followed by proteins identified in fewer than three replicates of each group. This filtered the total proteins from 1452 down to 654, which is the set upon which we performed our statistical analysis.

For the analysis of untreated drones and workers, newly emerged bees were marked with paint (Posca) and allowed to age in their respective hives for five days. They were then retrieved and hemolymph was immediately extracted using glass capillaries. The purpose of this experiment is to determine if the differential expression observed between drones and workers in the cage trials could simply be an artifact of being caged. Here, fresh hemolymph from uncaged drones and workers was collected and compared to determine if the same differential expression patterns are observed as between the control drones and workers in the prior cage trial. The hemolymph was prepared for proteomics exactly as described above, except 200 ng of peptides were analyzed on a Bruker TIMS-TOF mass spectrometer, and MaxQuant version 1.6.8.0 was used to process the data. Altered MaxQuant search parameters include: The precursor mass error for the main search was set to 50 ppm, and TIMS-DDA was selected within the "Type" tab of Group-Specific Parameters, and LFQ and match between runs were enabled. The identified proteins were then filtered exactly as described above.

**In-hive pesticide treatments**. Colonies were exposed to a pesticide cocktail via pollen patty feeding. Honey bees do not normally store pollen patty supplement and loss of pollen patty mass is presumed to represent consumption. The pesticide treatment mixture was based on bee bread residue data from Traynor et al.[24]. This mixture contained similar components as the topical application described above, with the exception that carbaryl was included and 2,4-DMPF was excluded, but with different relative proportions, owing to differences in detections in bee bread versus wax. The cocktail added to bee bread is described in detail in Table 2. Treated pollen patties received the pesticide mixture, while control pollen patties did not receive any added chemical treatment other than an equal amount of solvent. All pesticides (≥98% purity) were purchased from Sigma Aldrich Inc. or Chem Service Inc. Wildflower pollen (Glorybee Foods Inc. Eugene, Oregon) was

**Table 2 Target concentrations of the pesticide cocktail used for topical applications and pollen feeding[a].**

| Pesticide | Mode of action | Group | Median wax detection (ppb)[b] | Pollen (target ppb) | 0.33x topical (target ppb) | 2x topical (target ppb) | 10x topical (target ppb) |
|---|---|---|---|---|---|---|---|
| Coumaphos | Acetylcholine esterase inhibitor | Acaricide | 943 | 3260 | 314.3 | 1886.0 | 9430.0 |
| tau-Fluvalinate | Sodium channel modulator | Acaricide | 4310 | 469 | 1436.7 | 8620.0 | 43100.0 |
| 2,4-DMPF[c] | Octopamine receptor agonist | Acaricide (metabolite) | 304 | - | 101.3 | 608.0 | 3040.0 |
| Chlorothalonil | Multisite activity | Fungicide | 361 | 26600 | 120.3 | 722.0 | 3610.0 |
| Chlorpyrifos | Acetylcholine esterase inhibitor | Insecticide | 2.7 | 33.4 | 0.9 | 5.4 | 27.0 |
| Fenpropathrin | Sodium channel modulator | Insecticide | 16.8 | 24.6 | 5.6 | 33.6 | 168.0 |
| Pendimethalin | Inhibition of microtubule assembly | Herbicide | 5.3 | 143 | 1.8 | 10.6 | 53.0 |
| Atrazine | Inhibition of photosynthesis | Herbicide | 5.4 | 37.3 | 1.8 | 10.8 | 54.0 |
| Azoxystrobin | Cytochrome bc1 inhibitor | Fungicide | 5.1 | 83.1 | 1.7 | 10.2 | 51.0 |
| Carbaryl | Acetylcholine esterase inhibitor | Insecticide | - | 364 | - | - | - |

[a]Some elements of the recipe have been previously published[29].
[b]Concentrations are based off wax residue data published by Traynor et al.[24].
[c]2,4-Dimethylphenyl formamide (Amitraz degradate).

**Table 3 Sample sizes for proteomics experiments.**

| Experiment | Group | Sex | # Samples | Colony sources[a] |
|---|---|---|---|---|
| Cage exposure | Imidacloprid (1 ppm) | Worker | 6 | Farm002 (2), SL NE (2), SL SE (2) |
| Cage exposure | Imidacloprid (1 ppm) | Drone | 7 | Farm002 (3), SL NE (2), SL SE (2) |
| Cage exposure | Cocktail (10 x) | Worker | 8 | Farm002 (3), SL NE (2), SL SE (3) |
| Cage exposure | Cocktail (10 x) | Drone | 8 | Farm002 (3), SL NE (2), SL SE (3) |
| Cage exposure | Acetone | Worker | 6 | Farm002 (2), SL NE (2), SL SE (2) |
| Cage exposure | Acetone | Drone | 8 | Farm002 (2), SL NE (2), SL SE (2) |
| No cage | Untreated | Worker | 14 | Roof002 (7), Roof004 (7) |
| No cage | Untreated | Drone | 14 | Roof002 (7), Roof004 (7) |

[a]Number of bees from each source in brackets.

powdered using a laboratory blender and mixed with a sucrose solution to create a pollen supplement (final concentrations: 43% pollen, 43% granulated sucrose, 14% water by mass). Acetone, which quickly evaporates at room temperature, was used as the solvent for pesticide dilutions and comprised < 2% of the final diet. Components were thoroughly mixed in a stainless-steel bowl using an electric hand mixer (Model 62633R, Hamilton Beach, Glenn Allen, VA) and 40 g portions were placed on individual wax paper sheets and sealed in plastic bags at −20 °C until use. We have obtained pollen from this supplier (Glorybee Foods Inc.) previously and background pesticide residue testing showed very low detections of a limited number of contaminants, which did not pose meaningful risk to the bees (see Milone et al. (2021)[30] and Milone and Tarpy (2021)[28], and methods therein).

This experiment was carried out at the Lake Wheeler Honey Bee Research Facility (Raleigh, NC). Six colonies housed in single-deep standard Langstroth hive boxes roughly equilibrated for initial colony conditions (e.g., worker population, pollen and honey stores, and capped and uncapped brood) were fitted with pollen traps to encourage consumption of control and treated pollen patties. These colonies were continually exposed by placing a treated or untreated pollen patty on the top bars of each colony and consumption was recorded daily.

**Adult drone fostering experiment.** This experiment was conducted in April–May with all drones sampled for analyses from May 3rd to May 7th. Frames of emerging drone brood were selected from two source colonies and allowed to emerge for 5 d in an incubator set to 33 °C and 55% relative humidity. Emerged drones were collected daily in the morning and marked according to their source, placed into cages constructed of wood and queen excluders, and installed in one of six foster colonies by placing them atop the central frames of the hive using a 3 cm wooden spacer[54]. Foster colonies were treated with pollen patties as described above (three control, three treated). Foster colonies were fed either pesticide or control pollen patties for at least 28 d prior to the experiment. After 12 d had passed, drone cages were pulled from the colonies, the survivors were tallied, and mean mass was taken. A subset of 20 drones from each cage were individually sampled for morphometric and reproductive measures.

**Drone larval rearing experiment.** This experiment was conducted in May–June. Empty drone frames were placed into six experimental colonies (three treated and three control) once they had been fed control or treated pollen patties for at least 28 d (depending on queen laying time maximum 32 d). The six colonies used for drone larval rearing were the same colonies used to foster adult drones earlier in the season. Only two of the pesticide-treated colonies successfully reared drones. When the frames were capped and drone pupae had advanced to near-emergence (staged based on pupal eye colour being dark purple to black[55,56]), the frames were removed to an incubator set to 33 °C and 55% relative humidity for emergence. Bees were removed from each colony source daily, marked, and stocked into cages. All cages were fostered in a single, separate, untreated colony for 13 d post-emergence, when they were removed from the colony and immediately returned to the laboratory for dissection and analysis. The foster colony in this experiment did not receive pollen supplement treatment and did not have a pollen trap. Drones from multiple emergence days and colony sources (but not treatments) were combined into single cages and paint-marked, with 7–73 drones contained per cage.

**Drone collection, dissection, and morphometric analysis.** Drone collection and dissection proceeded according to previously established protocols[54]. Briefly, drones were weighed to the nearest 0.1 mg and anesthetized. The head and thorax were photographed, then drones were dissected to obtain their mucus glands. The seminal vesicles were cut free from the testicles and ejaculatory duct, then photographed. Finally, the head, wings, and abdomen were cut free from the thorax and legs and these were weighed. For experiment 1, the wings were not first removed from the thorax prior to weighing, therefore a corrective factor of 0.98 mg (the mean ($N = 5$) of the four wings) was subtracted from their thorax masses for further analyses. The seminal vesicles were ruptured in 1.0 mL Buffer D (3.0 g/L D-glucose, 4.1 g/L KCL, 2.1 g/L NaHCO₃, and 24.3 g/L Na₃C₆H₅O₇) and lightly

homogenized as previously described[54,57,58]. Live and dead spermatozoa were visualized using the Invitrogen live/dead spermatozoa staining kit #L7011 (Carlsbad, CA) and read using a Nexcelom Cellometer® Vision Sperm Counter machine (Nexcelom Bioscience LLC; Lawrence, MA, USA) to gain a count of spermatozoa and viable proportion. The photographs were then analyzed using ImageJ version 1.51m9[59] to measure the width of the head, thorax (as measured by the distance between tegulae), and the mean lengths of the seminal vesicles and mucus glands.

We defined body size as the first principal component of body mass, thorax mass, head width, and thorax width, and we defined fertility as the first principal component of total sperm count, sperm viability (arcsine-transformed proportion), mean seminal vesicle length, and mean mucus gland length. These principal components represented 67.8% of the variation in their loading variables for body size and 40.9% of their loading variables for fertility. We used these principal components to build linear models comparing the effects of colony source (a proxy for the combined effects of genetics and larval rearing environment) and adult rearing environment.

**Statistical analysis.** Unless otherwise stated, all statistical analyses were conducted using two-sided tests.

Adult drone topical- and cold-exposure survival counts were evaluated by logistic regression in R (v 3.6.0)[60]. Drones and workers were analyzed separately for each stressor using the generalized linear model with a binomial distribution where 1 was defined as "survived" and 0 defined as "perished" and exposure was a continuous variable of either dose (for pesticide treatments) or duration of exposure (for cold treatments). Each stressor was analyzed separately. No colony source effect was identified for any stressor; therefore, this parameter was dropped from the model. The glm procedure was performed using the base-R stats package[60].

All statistical analyses on the proteomics data were conducted in Perseus[61] version 1.6.1.1 using simple two-group comparisons and a Benjamini–Hochberg correction, which is better suited for high-throughput analyses of gene or protein expression than Bonferroni, at 5% FDR. We did not have sufficient statistical power to test for colony source effects for the cage trial data; however, we included samples with roughly equal representation of each colony so as not to bias the analysis (see Table 3). For the uncaged worker and drone comparison, we were able to test for source effects between the two colonies sampled and none were identified, so this factor was not included in our statistical model.

All analyses related to the hive exposures were performed in R (version 4.0.2), using $\chi^2$ tests for mortality. Statistical analyses were conducted using linear mixed models, with parameter significances reported as a $t$ distribution with estimated degrees of freedom or as a likelihood ratio test of the model with the parameter removed reported as a $\chi^2$. These analyses were performed using the lme4 package[62]. For experiment 1, where adults from two sources were installed into multiple treatment or control bank colonies, drone source colony and treatment were considered fixed effects and bank colony was considered a random, nested effect. For experiment 2, where larvae from control or treated colonies were reared to adulthood and banked in a common colony, treatment was considered a fixed effect and source a random effect.

**General aspects of statistics, reproducibility, and animal ethics.** Investigators were not blinded, as the endpoints measured were either binary (dead/alive) or not sensitive to user bias (proteomic sample preparation). No sample size calculation was performed to determine replicate numbers. We chose sample sizes which were consistent with our previous proteomic experiments investigating effect of stressors in honey bees[29]. Drones and workers were randomly allocated to their experimental groups, and sample orders were randomized ahead of LC-MSMS injection. Honey bees were sampled from colonies maintained using standard best management practices at the University of British Columbia and North Carolina State University. As non-cephalopod invertebrates, honey bees are not subject to Animal Ethics oversight at the University of British Columbia and North Carolina State University.

**Reporting summary**. Further information on research design is available in the Nature Research Reporting Summary linked to this article.

## Data availability

The data depicted by the figures are contained within Supplementary Data 1–6. All raw mass spectrometry data are available on MassIVE (www.massive.ucsd.edu/ProteoSAFe/static/massive.jsp, accession: MSV000087818)[53].

## Code availability

Any R codes associated with this manuscript are available from the authors upon request. Previous versions of R are available at https://cran.r-project.org/bin/windows/base/old/.

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

## Acknowledgements

We would like to acknowledge Jason Rogalski and Renata Moravcova for running the mass spectrometry instruments, and Bradford Vinson and Jennifer Keller for helping to maintain the honey bee colonies. This work was supported by Genome Canada (264PRO), Genome BC, and BC Ministry of Agriculture grants to L.J.F. and a Project Apis m grant and Boone Hodgson Wilkinson Trust Fund grant to A.M., L.J.F., and D.R.T. AM's salary was supported by a fellowship from the Natural Sciences and Engineering Research Council of Canada. Funding support for BNM was provided by the USDA-NIFA project 2016-07962 and grant W911NF1920306 from the USARL. J.P.M. was funded by a Graduate Student Fellowship through the NC Agricultural Foundation, a grant from the Foundation for Food and Agriculture Research (Grant #549053), and by the Foundation for the Preservation of Honey Bees.

## Author contributions

A.M. conducted the topical pesticide and cold exposure experiments, proteomics data acquisition, statistical analyses, and interpretation. J.M. and B.M. conducted the hive treatment experiments. B.M. conducted statistical analyses and interpretation of hive treatment experiments. J.M. produced the pesticide cocktails for topical and pollen exposures. Grants to L.J.F., D.R.T., and J.M. funded the research. A.M. wrote the first draft of the manuscript, with assistance from B.M. All authors edited and approved the final version of the manuscript and contributed to the work intellectually.

## Competing interests

The authors declare no competing interests.
