## [Peer Review File · Communications Biology]

Reviewers' comments:

Reviewer #1 (Remarks to the Author):

Bees comprise the most important pollinators. Honey bees are widely distributed around the globe and play pivotal roles in pollinating wild and cultivated plants and provide different commercial products such as honey, propolis, pollen, and wax. These bees were domesticated a long time ago and their management depends on the quality of drones. Pesticide poisoning is considered one of the causes of bee decline. Much of what is known about the sublethal effects of agrochemicals on bees was studied in workers, not drones. In this manuscript, it was studied the direct and indirect effects of the exposure of bees (drones and workers) to imidacloprid (neonicotinoid) and a mixture with different agrichemicals with different spectra of action. These agrichemicals include contaminants commonly found within the colony. The expression profiles of proteins of the hemolymph were generated and compared in exposed individuals with control (solvent) and individuals from different sexes (workers x drones). The fertility (not fecundity) and phenotype of treated and untreated groups were also compared. I'll point some issues that in my view will improve the manuscript.

Drones are seasonal and die after fertilization. They are costly to the colony, not performing crucial tasks in the colony such as nurses or foragers. Would they be expected to invest or be able to react metabolically to abiotic stresses such as workers?

To confirm the immune changes after the exposure, it should be performed an assay to test the immune response. A simple assay with nylon or bacterium injection should cover this gap.

The fertility was evaluated, not fecundity (L155, 357).

How many individuals were used in the assays? Please clarify the methods.

Explain why it was used imidacloprid. It is a neurotoxic insecticide. Why it is expected to change immune parameters?

Collecting hemolymph from thawed insects was not a good thing to do (necrotic tissue from cadavers). The freezing process (-20) destroys the tissues and cell content from different organs leaks and contaminates the hemolymph. Then the protein profile did not correspond to hemolymph solely. Hemolymph is very sensitive and many reactions involving protein modifications, including precipitation, coagulation, take place during the death, which hampered protein profiling and consequently, data interpretation. Fresh hemolymph should be used for all proteomics assays to avoid artifacts.

It was not mentioned if the exposure was chronic or acute. There are some works with chronic exposure considering agrichemical concentration within the colony (more "realistic") that could help in the discussion (e.g. DOI: 10.1016/j.envpol.2019.113420).

Minors:

L65: '...accumulate in various hive matrixes...' ok, but in a lower concentration than in the field?

Fig. 1. What do the numbers in the dots mean? Please, explain in the legend.

Fig. 2. Color the bars of the graphics in such a way that it is possible to better differentiate the colors and not to use color gradients.

L383: you mean: active ingredient?

Table 1: provide the brand of compounds.

Methods: what is the paint (type) used to mark the bees?

Reviewer #2 (Remarks to the Author):

Drone honey bees (*Apis mellifera*) are disproportionately sensitive to abiotic stressors despite expressing high levels of stress response proteins

Alison McAfee*^{1,2,3}, Bradley Metz*^{1,2}, Joseph P Milone¹, Leonard J Foster³, David R Tarpy^{1,2}

This paper investigates the proteomics response of drones to stressors such as imidacloprid and a general toxin cocktail. Survival was assessed additionally for temperature stressed drones. Both were compared to the response in workers. However the response of workers was not assessed along, although this is not completely clear. A major concern with this study is the small sample

size. It appears that the authors assessed individual drones kept in cages alone with only a few nurse bees. This seems unusual and potentially flawed as nurse bees are unlikely to feed single drones.

The exposure to the pesticide was for a short period of time only, which may have led to a lower response than hypothesised. Generally the study is interesting however as outlined in comments below, the authors miss to discuss important literature around drones' responses to stressors. This may have changed their hypothesis, as it is well known that drones are more susceptible to stressors and recent reports that younger individuals express immune genes differentially to adult bees, which is unrelated to an immune response.

Line 23: this is not surprising (see below) as more sensitivity has been reported previously and detected in this study. The elevation of stress proteins can be a sign of accumulated stress and not necessarily an effective immune response.

The abstract here does not reflect the results as the results for workers exposed to imidacloprid are not reported, as far as we could tell.

Line 39 you are missing important references describing response of drones to stressors such as pesticides and disease. Eg. Fisher 2018, Kairo 2016, Chmiel 2020, Grassl 2018, Tison 2016; Harwood 2020

Possibly, reading these references the authors may change the overall message of the introduction as it is evident that there is a large amount of literature investigating the response of males. Generally the perception is that males are more susceptible and use their immune responses differently to workers.

Line 62: again you are missing important references

Results:

L82 – Please add an introduction to the results considering the topical treatments, experiment 1 and 2 as well as the untreated samples; so it is easier for the reader to navigate the data.

Line 102: this caption doesn't seem to match the figure. Figure 1 does not have a panel d). What are the numbers above each data point? Why are they so variable? I suggest to label the figure better and then also describe better in the caption. a) seems to be cut off on the right.

124 – were they the same colonies so in figure 2a is the first bar in control from the same colony as the first bar in the treatment? The statistics need to be nested.

129 – again if the control and treatment drones came from different colonies, this needs to be nested in the model. Colony source can result in a lot of differences in immune response and survival.

148 why are there only 2 treatment sources in Figure 2d? For treatment source B, only a very small number of bees were analysed. Is this a valid statistical treatment group?

176 – The conclusion that the lower levels of the toxic cocktail are not hazardous to drones in general maybe be a bit strong here. Maybe under these specific conditions, where they are not exposed to other stressors that may act synergistically with the pesticide cocktail. Further, this study assesses exposure after a short period only, the effects could take place later in life. This should be discussed.

178 – There were NO proteins significantly changed between workers treated with acetone and imidacloprid? If this is true, please clarify in the results.

191 – again, was the effect on workers tested? The figures only show the comparison drone imid – acetone and drone-worker.

198 – this is in control drones compared to control workers or treated drones vs treated workers? Please specify.

206 please rephrase - HSP70 Ab 206 (NP_001153544.1) was differentially expressed in both sex and pesticide exposure comparisons, but, 207 like p(2)el (XP_006568238.2; another HSP), expression increased with imidacloprid treatment for 208 drones, but not workers (Figure 4d). The sentence seems to be contradictory.

Line 306 – Upregulation of immune genes does not necessarily correlate with immune function, see

Bull JC, Ryabov EV, Prince G, et al. A strong immune response in young adult honeybees masks their increased susceptibility to infection compared to older bees. *PLoS Pathog.*

2012;8(12):e1003083. doi:10.1371/journal.ppat.1003083.

References and discussions to important literature on drones and neonicotinoids as well as other

stressors are missing.

Methods:

general: the methods need some clarification. the sample sizes in each of the experiments are not obvious or the origin of the test bees. A diagram of the experiments, source colony and host colonies, including samples sizes, cage, etc would be helpful.

375 – What is the genetic relatedness of each hive? Sister queens? How old were the queens? What is the disease and health status of each colony? What was the size of each colony?

377 – What is the size of the cages and how many bees are ideally placed in these? Provide a reference that drones housed on their own, even with workers will survive. From COLloss and our experience, workers will only feed drones when there is a substantial number.

377 – what is the sample size of each treatment?

392 – what was the humidity and where were they kept? Optimal conditions are important for survival. While all bees were treated the same, adding a second stressor may increase mortality.

418 - please include the MS running parameters

419 – please include the proteome database, including date and search parameters used.

426 – the untreated bees were treated differently, in that they were not frozen prior to haemolymph extraction. Has effects of this variation to the protocol been tested? The variation may introduced large artifacts in protein abundance due to cell breakage during the freeze-thaw cycle. Please consider and discuss.

427 – how many ng of protein were digested and ng of peptides analysed for the treatment groups?

428 – what is the purpose of using a different version of MAXQuant for the treatments and the untreated? Were these samples not compared to the treatments? Why were not all samples run on the same version together?

447 – how strong were the hives (number of brood frames, foundation) ? and how often were the patties replaced? Were the patties eaten in this period of time?

455 – were the colonies from sister queens?

Line 465 – were these the same colonies that the adult fostering experiment took place in? If so can this please be made clearer?

472 - 7-73 drones seems to be large range, please comment. We know that workers are more likely to feed drones when they are in large enough numbers in the cages, did the authors perform correlation analyses between drone number per cage and survival?

482 – Please include components of Buffer D

503- 505 – was the original colony (mother of drones and workers) included in the statistical model? This needs to be clearer

The Discussion seems feasible, although I would have liked to see a greater discussion of the different protein groups between workers and drones. Finally a greater discussion of the implications of neonicotinoid treatment on drones for colony strengths and future bee survival. Similar to the introduction many papers are not referenced and inclusion of these papers may allow a more interesting discussion.

Reviewer #3 (Remarks to the Author):

This paper is well written (with some minor issues that I have pointed out below) and easy to comprehend. It studies a topic of interest to me: the relative ability of haploid drones and diploid worker honey bees to respond to and survive stresses. The data seem strong and the analyses appropriate. (My background is not in proteomics so I am not well qualified to comment on that aspect of the research.) I feel this paper adds considerably to our understanding of this topic.

The 3 things that concern me the most that the authors should address, in my opinion, are:

1) The research is framed in the context of the "haploid susceptibility hypothesis". McAfee et al. in this paper framed that argument-- I believe incorrectly-- in terms of "deleterious recessive alleles".

This hypothesis is described relatively briefly in the 2nd paragraph of the Introduction, and again at the beginning of the Discussion, but if a main goal of the experiments was to test this

hypothesis, I think it should be addressed in somewhat more detail.

2) A number of recent studies have demonstrated that pollen collected by honey bees almost always contains residues of pesticides. In a 2007 study by researchers at Penn State University wrote in the American Bee Journal (Frazier et al.), "In a total of 108 pollen samples analyzed, 46 different pesticides including six of their metabolites were identified." For this reason, I was surprised that there were no analyses of the pollen purchased to feed colonies or of the pollen stored in colonies rearing drones. I don't think this is serious for the study (all treatments had the same background level of contaminants in pollen), but this should be addressed.

3. The discussion seems quite speculative to me. This may be normal for proteomics research, where the functions of proteins are assumed based on other research. I would prefer to see the amount of speculation reduced.

Below are my more detailed comments. The first one adds to #1 above.

Point by point reviewer response

Reviewer #1 (Remarks to the Author):

Bees comprise the most important pollinators. Honey bees are widely distributed around the globe and play pivotal roles in pollinating wild and cultivated plants and provide different commercial products such as honey, propolis, pollen, and wax. These bees were domesticated a long time ago and their management depends on the quality of drones. Pesticide poisoning is considered one of the causes of bee decline. Much of what is known about the sublethal effects of agrochemicals on bees was studied in workers, not drones. In this manuscript, it was studied the direct and indirect effects of the exposure of bees (drones and workers) to imidacloprid (neonicotinoid) and a mixture with different agrichemicals with different spectra of action. These agrichemicals include contaminants commonly found within the colony. The expression profiles of proteins of the hemolymph were generated and compared in exposed individuals with control (solvent) and individuals from different sexes (workers x drones). The fertility (not fecundity) and phenotype of treated and untreated groups were also compared. I'll point some issues that in my view will improve the manuscript.

Thank you for providing this thoughtful feedback. As a result of your comments, we have made many clarifications to the manuscript (including changing fecundity to fertility) and have better explained our expectations and rationale for the experiments, which we hope will make more sense now. Please find the detailed responses below:

1. Drones are seasonal and die after fertilization. They are costly to the colony, not performing crucial tasks in the colony such as nurses or foragers. Would they be expected to invest or be able to react metabolically to abiotic stresses such as workers?

This is a good point that warrants clarification. In honey bees, natural selection would act directly on drones and queens (the reproductives) and indirectly on workers. So, we suggest that although drones do not execute critical colony tasks like foraging and nursing, but do experience pesticide residues in the hive and execute the most critical task of all for species survival (reproduction), they should theoretically experience some selection for stress tolerance despite being shielded from some extremes that workers may experience. Indeed, queens, which are similarly shielded from exterior environment and are similarly invested in reproduction, are generally tolerant of stress. In addition, in our previous research we have found that the hive periphery, where drones tend to congregate, can undergo dramatic temperature fluctuations (see McAfee et al. *Nature Sustainability*). We have added the following clarifying text to the introduction (line 68):

“Since drones do not forage, they are most likely to encounter more complex pesticide mixtures that accumulate in various hive matrixes (e.g., wax, pollen, honey). Furthermore, because drones encounter contaminated wax, consume pollen and honey, and experience temperature variation in the hive and on mating flights, we expect that they, like workers, experience some selection for tolerance mechanisms. They also experience indirect selection through their sister workers (50% relatedness) and mother queen (100% relatedness), whose genes they share.”

2. To confirm the immune changes after the exposure, it should be performed an assay to test

the immune response. A simple assay with nylon or bacterium injection should cover this gap. The fertility was evaluated, not fecundity (L155, 357).

This, and the comments of the other reviewers, made us re-evaluate why such an emphasis was interpreted on immunity, when we were not actually investigating immunocompetence or disease challenges in this work. We assume this was because we have framed the work in terms of the haploid susceptibility hypothesis, which as proposed by O'Donnell and Beshers was originally addressing susceptibility to parasites and pathogens. Although the theoretical arguments should apply equally to biotic as well as abiotic stressors, we can find no prior mention of the haploid susceptibility hypothesis in the context of abiotic stress. We have clarified this observation and added that we propose the hypothesis should be expanded to include abiotic stress in light of the data we present as well as existing data (line 47):

“This apparent biased sensitivity of male bees, which are haploid, to abiotic stressors may be in part explained by an extension of the haploid susceptibility hypothesis, which states that haploid individuals are more susceptible to pathogenic infections, since they have no opportunity for allelic diversity that comes with heterozygosity¹⁵. O'Donnell and Beshers, who proposed the haploid susceptibility hypothesis¹⁵, described the notion as a corollary to the well-known heterozygous advantage¹⁶. While existing examples of haploid susceptibility are described in the context of disease and parasites, in theory the notion should equally apply to abiotic stressors.

*The haploid susceptibility hypothesis is not consistently supported when it comes to pathogenic infections, and has been challenged¹⁷. While investigations on honey bee male susceptibility to *Nosema*¹⁸, as well as immunocompetence of leafcutter ants (*Atta colombica*)¹⁹, wood ants (*Formica exsecta*)²⁰, and buff-tailed bumble bees (*Bombus terrestris*)²¹ support the haploid susceptibility hypothesis, research on *B. terrestris* male susceptibility to *Crithidia* does not¹⁷. However, this hypothesis has generally not been discussed in the context of abiotic stress, despite also being a relevant challenge.”*

And at Line 90:

“Our data suggest that drones have surprisingly strong baseline expression of putative stress response proteins, contrary to our expectations, causing us to re-evaluate exactly why drones, but not workers, are so intolerant to abiotic stress.

Regardless of the mechanism, we suggest that, in light of male sensitivity to abiotic stressors in honey bees, as documented here, the haploid susceptibility hypothesis may be expanded to include abiotic stress in addition to pathogens and parasites. Experiments investigating sex biases in other social insects are needed to determine how generalizable these observations are.”

We reiterate that we were investigating drone- and worker-stress responses to pesticides, and not focusing on immunocompetence specifically. We document that drones express high constitutive levels of putative stress response proteins, some of which are canonical immune factors, but most of which are others such as detoxification enzymes or heat-shock proteins.

3. How many individuals were used in the assays? Please clarify the methods.

We apologize for the oversight. Upon completing the data reporting checklist, we noted that we were missing these key details. We corrected the manuscript and resubmitted immediately but apparently not before the manuscript had already been sent to at least one reviewer. This version of the manuscript has been updated to include sample sizes for all experiments clearly indicated in the text and/or figure legends.

4. Explain why it was used imidacloprid. It is a neurotoxic insecticide. Why it is expected to change immune parameters?

As described in point 2 above, we actually did not expect that the pesticide would alter immune protein expression specifically. Rather, our aim was to compare the global protein changes in drones and workers exposed to equal doses of two different pesticide treatments relative to controls. We have added clarification of the specific topics we were addressing in the introduction (line 86):

“Given that drones are generally considered more sensitive to pesticide exposure than workers, which our data corroborates, we used these data to address our predictions that 1) drones have lower constitutive expression of relevant stress response proteins, and 2) that workers, but not drones, elevate detoxification enzymes in response to exposure.”

5. Collecting hemolymph from thawed insects was not a good thing to do (necrotic tissue from cadavers). The freezing process (-20) destroys the tissues and cell content from different organs leaks and contaminates the hemolymph. Then the protein profile did not correspond to hemolymph solely. Hemolymph is very sensitive and many reactions involving protein modifications, including precipitation, coagulation, take place during the death, which hampered protein profiling and consequently, data interpretation. Fresh hemolymph should be used for all proteomics assays to avoid artifacts.

This is a valid point, and we agree that freezing the bees first was not ideal; we were quite limited in what we could do owing to laboratory closures in the spring of 2020. However, we argue that because all the bees were handled in the same way, and our arguments do not depend on proteins being expressed in the hemolymph specifically, comparisons with the appropriate controls, as we have done, still address our specific hypotheses.

Furthermore, in the follow up experiment (conducted in the summer of 2021), we collected fresh hemolymph from drones and workers taken straight from the hive. As described at lines 253 to 263, these data are remarkably consistent with the dataset obtained from the frozen drones and workers, so freezing does not appear to have a dramatic effect on the proteomics analysis. We did identify more proteins in the fresh hemolymph samples (frozen: 1,452 identified, 654 quantified; fresh: 2,090 identified, 988 quantified), but the fresh hemolymph experiment was also conducted on a Bruker TIMS-TOF mass spectrometer, which routinely achieves higher coverage than the Bruker Impact II Q-TOF mass spectrometer owing to the superior sensitivity achieved by ion

mobility separation. The TIMS-TOF experiment also had higher replication (n = 14, compared to n = 6-8), which also contributes to more identifications.

6. It was not mentioned if the exposure was chronic or acute. There are some works with chronic exposure considering agrichemical concentration within the colony (more “realistic”) that could help in the discussion (e.g. DOI: 10.1016/j.envpol.2019.113420).

As we describe in the methods, the pesticide treatments were applied as single, 2 µl topical applications and bees were sacrificed 2 days later (constituting an acute exposure). Likewise, cold stress was applied as a one-time exposure to 4 C for specified lengths of time. We have added that these are both acute exposures to the methods (Line 443).

7. Minors:

L65: ‘...accumulate in various hive matrixes...’ ok, but in a lower concentration than in the field?

The answer to this depends on the matrix in question and which pesticides. Bee bread, for example, would arguably contain residue amounts similar to the field, since it is primarily made up of pollen collected from the field. Wax, however, would be expected to contain lower concentrations of some pesticides and higher concentrations of others depending the compounds physiochemical properties. Since wax primarily acquires lipophilic hive products or beekeeper-applied miticides there is also the potential for residue magnification over repeated exposures/applications. In light of these distinctions, we think it is best not to discuss pesticide abundance in general terms of being higher or lower than in the field.

Fig. 1. What do the numbers in the dots mean? Please, explain in the legend.

Apologies, this is one of the omissions we noted while conducting the reporting checklist. The numbers indicate the exact sample size of individual drones tested for each condition, as is now indicated in the legend.

Fig. 2. Color the bars of the graphics in such a way that it is possible to better differentiate the colors and not to use color gradients.

Done. We have updated the figure:

Figure 2. No consistent effects of hive-level cocktail treatments via pollen on drone development or adult fostering. Drones were exposed to colony-level pollen-delivered pesticide cocktail either as adults (experiment 1) or as larvae (experiment 2). Unique drone source colonies are differentiated by color. Number of individuals included in each group is displayed over their respective bar or boxplot. Mortality data are reported as proportion dead presented in color, with live proportion presented in grey. Mortality differences were evaluated with χ^2 tests, whereas size and fertility were evaluated using linear mixed models (see methods for specific models). Boxes represent the interquartile range, bars indicate the median, and whiskers span 1.5 times the interquartile range. Sample sizes (n) are printed at the column break for the mortality tests (a & d) and along the top of each box plot (b-c & e-f). (a-c) Mortality, size, and fertility of drones from different source colonies reared in untreated colonies but fostered in either treated and untreated colonies as adults. (d-f) Mortality, size, and fertility of drones from different source colonies which were reared through development by treated and untreated colonies, but fostered in untreated colonies as adults.

L383: you mean: active ingredient?

Yes, that is correct. We reworded the sentence to clearly indicate this: “To analyze pesticide stress, bees were briefly anesthetized with carbon dioxide to immobilize them, then 2 μ l of pesticide active ingredient solution (either a cocktail mixture or imidacloprid, which was not part of the cocktail, in acetone) was applied directly to the thorax as an acute exposure.”

Table 1: provide the brand of compounds.

We have added the following to the methods (line 449):

“As previously described, all compounds in the pesticide mixture were purchased as pure technical material (\geq 95% purity) from Sigma Aldrich (St. Louis, MO) or Chem Service Inc. and were serially diluted in acetone in order to achieve the respective concentrations²⁸.”

Methods: what is the paint (type) used to mark the bees?

We have now indicated that Posca (Japan) paint pens were used (line 425).

Reviewer #2 (Remarks to the Author):

Drone honey bees (*Apis mellifera*) are disproportionately sensitive to abiotic stressors despite expressing high levels of stress response proteins

Alison McAfee*1,2,3, Bradley Metz*1,2, Joseph P Milone1, Leonard J Foster3, David R Tarpy1,2

1. This paper investigates the proteomics response of drones to stressors such as imidacloprid and a general toxin cocktail. Survival was assessed additionally for temperature stressed drones. Both were compared to the response in workers. However the response of workers was not assessed along, although this is not completely clear.

We appreciate the great lengths this reviewer has gone to in order to discuss the manuscript in detail. We think that the manuscript is much improved as a result, and discuss the individual points below. We will discuss this later as well but we note that the proteomic response of workers was in fact analyzed as well but as shown in Fig 3c, differential expression was not observed.

2. A major concern with this study is the small sample size. It appears that the authors assessed individual drones kept in cages alone with only a few nurse bees. This seem unusual and potentially flawed as nurse bees are unlikely to feed single drones.

We are unsure of exactly which part of the study the reviewer is addressing here, regarding sample sizes, but we assume it is the cage trial. As we have noted to Reviewer 1 above, the numbers above and below the dots in Figure 1 indicate the sample size (# of

drones or workers) tested for each condition, which ranges from 13 to 48 individuals from at least three different colonies. These sample sizes are well within the expected range of similar papers on this topic. This explanation is now included in the legend of Figure 1. The cold stress dataset has the fewest replicates, but given that a large effect was observed for both the 2 h and 4 h time points, the effect of exposure on drone survival was still highly significant and therefore not constrained by the sample size.

Regarding the cage design, we note that drone survival in the control groups was always very high, between 85 and 90%. Therefore, there was some baseline stress of the cage, which is in part why we repeated the proteomics experiment with drones and workers sampled directly from the colony. The baseline stress is somewhat expected – cage experiments are stressful even for workers, who do feed themselves. However, we note that these cages included fondant for food rather than syrup, which drones are able to eat. Drones will not drink syrup because they have no proboscis, but they are apparently able to chew fondant. During the course of the experiment, we observed both trophallaxis from workers to drones as well as self-feeding behavior of drones. We expect that the baseline survival of drones was slightly lower than workers owing to the inherent stress of the cage, rather than starvation, but reiterate that our final proteomics experiment was conducted without using cages (data in Fig 6).

3. The exposure the pesticide was for a short period of time only, which may have lead to a lower response than hypothesised. Generally the study is interesting however as outline in comments below, the authors miss to discuss important literature around drones responses to stressors. This may have changed their hypothesis, as it is well known that drones are more susceptible to stressors and recent reports that younger individuals express immune genes differentially to adult bees, which is unrelated to a immune response.

It is true that this experiment was conducted over a short period of time. While chronic, low-dose exposures are more informative for studying impacts of pesticides on bees under field conditions, our goal is actually not limited to understanding what happens in a managed scenario (although this is also addressed in our hive exposures). Rather, our goal was to investigate sex biases in stress responses more generally, which could be addressed by either chronic or acute exposures, and we opted for acute owing to the simpler methodology. We have clarified that these are our goals at the end of the introduction (line 80)

“Here, we aim to investigate drone and worker tolerances to abiotic stressors, focussing mainly on pesticide exposure. Our overarching goal was to investigate the response of putative stress response proteins that could potentially underly sex biases in tolerance.”

...

“Given that drones are generally considered more sensitive to pesticide exposure than workers, which our data corroborates, we used these data to address our predictions that 1) drones have lower constitutive expression of relevant stress response proteins, and 2) that workers, but not drones, elevate detoxification enzymes in response to exposure. Our data suggest that drones have surprisingly strong baseline expression of putative stress response proteins, contrary to our expectations, causing us to re-evaluate exactly why drones, but not workers, are so intolerant to abiotic stress.”

So, the additional references that identify drone sensitivity to different pesticides do not actually change our hypotheses but strengthen their premise. We always hypothesized that drones would be more sensitive than workers, and were most interested in what the proteomics data might reveal as a potential explanation.

Regarding the point about younger individuals expressing altered levels of immune genes compared to older individuals, we are not sure how that relates to our experiments exactly since we are not dealing with pathogenic infections and all our bees were age-matched.

4. Line 23: this is not surprising (see below) as more sensitivity has been reported previously and detected in this study. The elevation of stress proteins can be sign of accumulated stress and not necessary a effective immune response.

Thank you for pointing this out. As we describe below, we have updated the introduction to include reference to these additional relevant papers. The novelty of our manuscript is really that it couples the observed sex biases in tolerance to proteomic analyses, which address molecular processes that may underly these differences. As described in point 3 above, the findings are surprising, especially in light of the second proteomic experiment (Figure 6) (which sampled hemolymph from drones and workers straight from their hives, without the added stress of the cage) corroborate the findings of the first experiment (Figure 3-5).

As we pointed out in our response to Reviewer 1 and point 3 above, these experiments are not investigating pathogens or immune responses specifically. We suspect that this assumption is being made because of how we framed our work in the context of haploid susceptibility, which originally was proposed as an explanation for male susceptibility to pathogens. We have added an explanation in the introduction addressing the idea that there is no theoretical reason why haploid susceptibility should not be relevant for abiotic stressors as well, and suggest that the scope of haploid susceptibility should be expanded to reflect this, especially in light of the accumulating data on sex biases and temperature and pesticide tolerance.

Line 47:

“This apparent biased sensitivity of male bees, which are haploid, to abiotic stressors may be in part explained by an extension of the haploid susceptibility hypothesis, which states that haploid individuals are more susceptible to pathogenic infections, since they have no opportunity for allelic diversity that comes with heterozygosity¹⁵. O’Donnell and Beshers, who proposed the haploid susceptibility hypothesis¹⁵, described the notion as a corollary to the well-known heterozygous advantage¹⁶. While existing examples of haploid susceptibility are described in the context of disease and parasites, in theory the notion should equally apply to abiotic stressors.

*The haploid susceptibility hypothesis is not consistently supported when it comes to pathogenic infections, and has been challenged¹⁷. While investigations on honey bee male susceptibility to *Nosema*¹⁸, as well as immunocompetence of leafcutter ants (*Atta colombica*)¹⁹, wood ants (*Formica exsecta*)²⁰, and buff-tailed bumble bees (*Bombus terrestris*)²¹ support the haploid susceptibility hypothesis, research on *B. terrestris* male*

susceptibility to Crithidia does not¹⁷. However, this hypothesis has generally not been discussed in the context of abiotic stress, despite also being a relevant challenge.

And at Line 90:

“Our data suggest that drones have surprisingly strong baseline expression of putative stress response proteins, contrary to our expectations, causing us to re-evaluate exactly why drones, but not workers, are so intolerant to abiotic stress.

Regardless of the mechanism, we suggest that, in light of male sensitivity to abiotic stressors in honey bees, as documented here, the haploid susceptibility hypothesis may be expanded to include abiotic stress in addition to pathogens and parasites.

Experiments investigating sex biases in other social insects are needed to determine how generalizable these observations are.”

5. The abstract here does not reflect the results as the results for worker exposed to imidacloprid are not reported, as far as we could tell.

This is true, we did not report those results in the abstract because we did not find any differences there and wanted to focus on the significant results we did identify. Lack of significance is still a result, though, so we have now included this description in the abstract:

“We then used quantitative proteomics to investigate protein expression profiles in the hemolymph of topically exposed workers and drones, and found that 34 proteins were differentially expressed in exposed drones relative to controls, but none were differentially expressed in exposed workers.”

6. Line 39 you are missing important references describing response of drones to stressors such as pesticides and disease. Eg. Fisher 2018, Kairo 2016, Chmiel 2020, Grassl 2018 , Tison 2016; Harwood 2020. Possibly, reading these references the authors may change the overall message of the introduction as it is evident that there is a large amount of literature investigating the response of males. Generally the perception is that males are more susceptible and use their immune responses differently to workers.

Thank you for pointing out these papers. The Fisher and Kairo papers investigate the effect of pesticide mixtures or fipronil, respectively, on drone fertility, and have been added to the relevant section in the introduction, but survival of workers and drones were not comparable because they either were not tested or they were not treated equally (this was not an original goal of the papers). The Tison paper and Harwood paper do not address effects on drones specifically. The Chmiel paper is a review not focusing on drones, and we think the Rangel review on drone stress tolerance we cited is more appropriate, but have added reference to the Chmiel paper for a discussion on worker responses to pesticides. The Grassl paper does indicate poor survival of drones exposed to thiamethoxam compared to workers and has been added.

Indeed, this body of work, along with others we already cited, supports the notion of male susceptibility. As stated above, our hypothesis was that males would be more

susceptible to the stressors we tested as well, so this aspect remains the same. Our primary interest was in the underlying proteomic shifts, and in making the argument to consider generalizing the haploid susceptibility hypothesis to include abiotic stressors.

7. Line 62: again you are missing important references

Please see above – these references have now been added to the relevant sections. A reference to the Grassl paper has also been added to the beginning of the results.

Results:

8. L82 – Please add a introduction to the results considering the topical treatments, experiment 1 and 2 as well as the untreated samples; so it is easier for the reader to navigate the data.

Thank you for this suggestion. We have indicated that the topical exposures were applied in 2 µl of acetone, the doses tested, and included the duration and temperature of cold exposures. We have also indicated that survival was measured 2 days after exposure and that these were cage experiments. More detail than this is probably too many methods, but these details should help clarify the context.

Regarding the hive experiments, we have added the following text (line 137):

“In the first experiment, we banked naïve adult drones either in colonies that had been previously fed a pesticide pollen patty supplement for 28 d, or in colonies that were fed pollen patties with no added pesticides.”

And this explanation to line 158:

“We first fed colonies either pesticide or control pollen patties for 28 d, then inserted empty drone frames to be laid out and continued to feed the patties throughout subsequent drone development.”

9. Line 102: this caption doesn't seem to match the figure. Figure 1 does not have a panel d). What are the numbers above each data points? Why are they so variable? I suggest to label the figure better and then also describe better in the caption. a) seems to be cut off on the right.

Apologies for the oversight. We had initially presented the data in four panels, with baseline survival rates in panel a and survival rates of the stress treatments normalized to baseline in b-d. However, we decided it was clearer to present the baseline (control) survival rate on the same graph and not to normalize the values, so there is one less panel as well. We neglected to update the text and legend to reflect this, but have done so now. We note that we also neglected to update the description of the statistics in the legend, which still indicated a chi square test, when we had updated it to a generalized linear model with binomial distribution.

10. 124 – were they the same colonies so in figure 2a is the first bar in control from the same colony as the fist bar in the treatment? The statistics needs to be nested.

We did not use a paired design in this experiment (where a colony was measured before and after the treatment). As discussed in the methods section (Line 527), Colonies were independent replicates and were either received treated or untreated pollen patties. This methodology was selected to mitigate seasonal effects and also to maximize resources.

11. 129 – again if the control and treatment drones came from different colonies, this needs to be nested in the model. Colony source can result in a lot of differences in immune response and survival.

The authors agree with the reviewer that genetic and colonial variation can influence responses to stressors. The way we implemented model nesting was stated in the methods section at line 600.

“For experiment 1, where adults from two sources were installed into multiple treatment or control bank colonies, drone source colony and treatment were considered fixed effects and bank colony was considered a random, nested effect. For experiment 2, where larvae from control or treated colonies were reared to adulthood and banked in a common colony, treatment was considered a fixed effect and source a random effect.”

12. 148 why are there only 2 treatment sources in Figure 2d? For treatment source B, only a very small number of bees were analysed. Is this a valid statistical treatment group?

During the experiment, one of the pesticide-treated colonies was weakened to the point where it could no longer be used as a source of drones. As a result, we had to exclude it from the experiment since it was not a valid test of the hypothesis. We state that a colony failed to produce drones on line 162 and at line 549. Increased replication would have improved our statistical power but we feel that our conclusions were adequately supported by our current dataset.

13. 176 – The conclusion that the lower levels of the toxic cocktail are not hazardous to drones in general maybe be a bit strong here. Maybe under these specific conditions, where they are not exposed to other stressors that may act synergistically with the pesticide cocktail. Further, this study assesses exposure after a short period only, the effects could take place later in life. This should be discussed.

We agree that this conclusion should be tempered. We have added the following text to the results (line 199) and discussion (line 407):

“No differences were identified in any of the other pairwise comparisons, including cocktail treatments relative to controls, as well as all exposed worker comparisons, further supporting that the cocktail is not hazardous to drones at these doses and under these conditions.”

“We note that different results may be obtained for bees from a different genetic stock or which are experiencing combined effects of other stressors.”

14. 178 – There was NO proteins significantly changed between worker treated with acetone and imidacloprid? If this is true, please clarify in the results.

Correct. We have updated the text to read (line 199): “No differences were identified in any of the other pairwise comparisons, including cocktail treatments relative to controls, as well as all exposed worker comparisons.”

15. 191 – again, was the effect on workers tested? The figures only show the comparison drone imid – acetone and drone-worker.

Yes, the effect on workers was tested. Figure 3c shows a summary of differentially expressed proteins identified in different pairwise comparisons, indicating 188 proteins differentially expressed between control drones and workers, 34 differentially expressed proteins between imidacloprid exposed drones and control drones, and 0 differentially expressed proteins for all other comparisons.

16. 198 – this is in control drones compared to control workers or treated drones vs treated workers? Please specify.

Thank you for pointing this out. We are referring to control drones compared to control workers, and have now indicated this in the text.

17. 206 please rephrase - HSP70 Ab 206 (NP_001153544.1) was differentially expressed in both sex and pesticide exposure comparisons, but, 207 like pl(2)el (XP_006568238.2; another HSP), expression increased with imidacloprid treatment for 208 drones, but not workers (Figure 4d). 18. The sentence seems to be contradictory.

Apologies for the poor wording. We have updated the text (line 231):

“HSP70 Ab (NP_001153544.1) was differentially expressed in both sex and pesticide exposure comparisons. Expression of the small heat-shock protein pl(2)el (XP_006568238.2), however, increased with imidacloprid treatment for drones, but not workers (Figure 4d)”

19. Line 306 – Upregulation of immune genes does not necessarily correlate with immune function, see 20. Bull JC, Ryabov EV, Prince G, et al. A strong immune response in young adult honeybees masks their increased susceptibility to infection compared to older bees. PLoS Pathog. 2012;8(12):e1003083. doi:10.1371/journal.ppat.1003083. References and discussions to important literature on drones and neonicotinoids as well as other stressors are missing.

This is a good point. We have updated the text to specify *immune protein expression* and not immune function, which we did not test. But, to the larger point of what is brought up in the Bull et al. paper, their observations appear to support the argument that high constitutive expression of immune factors is a better indicator of resistance than immune inducibility. This argument is in line with our guiding expectations, that drones’

apparent fragility could be explained by either 1) low baseline expression of stress response proteins relative to workers, or 2) insufficient inducibility of their response. However, our results are not consistent with the idea that high baseline expression coincides with greater resistance to stress, since drones are highly sensitive but express high levels of stress response proteins constitutively. We have added the following text to the discussion to address our data in the context of Bull et al.'s work (Line 344):

“These results also raise the question of how workers are so stress tolerant, in terms of survival, without launching an equally robust stress response. Indeed, we identified no differentially expressed proteins comparing workers treated with imidacloprid to controls. While investigating worker responses to an entomopathogenic fungus, Bull et al. suggest that immune inducibility is not a reliable indicator of resistance, and it is possible that the same may be true with regard to abiotic stress responses. However, Bull et al. also propose that higher baseline expression of immune proteins in foragers relative to workers may explain why foragers are more resistant, despite exhibiting low inducibility. This creates a conundrum, because workers are generally more stress tolerant, and we identified both low inducibility and low baseline expression of stress response proteins relative to drones. On the whole, these data indicate that differential expression is difficult to interpret in the absence of protein activity, but the question remains: Why do drones express such high levels of putative stress response proteins, if they ostensibly remain inactive?”

Methods:

21. general: the methods need some clarification. the sample sizes in each of the experiments are not obvious or the origin of the test bees. A diagram of the experiments, source colony and host colonies, including samples sizes, cage, etc would be helpful.

We have included colony source information for all experiments in the supplemental tables, as well as a new table in the methods section (line 439):

Table 1. Sample sizes and colony origins for drone and worker survival comparisons.

Stressor	Sex	Colony	# Bees
Imidacloprid	Worker	Farm 002	26
Imidacloprid	Drone	Farm 002	30
Imidacloprid	Worker	Roof 002	20
Imidacloprid	Drone	Roof 002	13
Imidacloprid	Worker	Roof 004	19
Imidacloprid	Drone	Roof 004	13
Imidacloprid	Worker	SL NE	25
Imidacloprid	Drone	SL NE	32
Imidacloprid	Worker	SL SE	27
Imidacloprid	Drone	SL SE	29
Cocktail	Worker	Farm 002	26
Cocktail	Drone	Farm 002	31
Cocktail	Worker	SL NE	22
Cocktail	Drone	SL NE	26
Cocktail	Worker	SL SE	30
Cocktail	Drone	SL SE	26

Cold	Worker	Farm 002	17
Cold	Drone	Farm 002	18
Cold	Worker	SL NE	15
Cold	Drone	SL NE	16
Cold	Worker	SL SE	18
Cold	Drone	SL SE	13

We have indicated in the legend of figure 1 that the numbers above and below the dots indicate sample sizes. We have also indicated that the numbers at the top of the boxplots and at the column breaks on the proportion dead plots represent sample sizes. The following table is for the reviewer's reference only (the information is already presented within Figure 2, where n for each group is indicated).

		Drone Foster Colonies						
Drone Sources		1	2	3	4	5	6	Z
		Control	Control	Control	Treatment	Treatment	Treatment	Control
1	Control	20	20	20	20	21	20	
2	Control	19	20	21	20	22	20	
A	Control							27
B	Control							19
C	Control							37
D	Treatment							69
E	Treatment							13

Proteomics sample sizes are now included in Table 3 (line 510):

Table 3. Sample sizes for proteomics experiments.

Experiment	Group	Sex	# samples	Colony sources*
Cage exposure	Imidacloprid (1 ppm)	Worker	6	Farm002 (2), SL NE (2), SL SE (2)
Cage exposure	Imidacloprid (1 ppm)	Drone	7	Farm002 (3), SL NE (2), SL SE (2)
Cage exposure	Cocktail (10 x)	Worker	8	Farm002 (3), SL NE (2), SL SE (3)
Cage exposure	Cocktail (10 x)	Drone	8	Farm002 (3), SL NE (2), SL SE (3)
Cage exposure	Acetone	Worker	6	Farm002 (2), SL NE (2), SL SE (2)
Cage exposure	Acetone	Drone	8	Farm002 (2), SL NE (2), SL SE (2)
No cage	Untreated	Worker	14	Roof002 (7), Roof004 (7)
No cage	Untreated	Drone	14	Roof002 (7), Roof004 (7)

*number of bees from each source in brackets

22. 375 – What is the genetic relatedness of each hive? Sister queens? How old were the

queens? What is the disease and health status of each colony? What was the size of each colony?

We have added the following explanation (line 433):

“Colonies were headed by genetically unrelated queens which were produced the summer prior to experimentation and had successfully overwintered. The colonies “Farm 002,” “SL NE,” and “SL SE” had two standard deep hive bodies whereas colonies “Roof 002” and “Roof 004” were single box colonies. All colonies were treated for Varroa mites using Apivar the previous fall, but spring mite treatments were delayed until after drones and workers were collected. Only bees free of Varroa at emergence were marked.”

23. 377 – What is the size of the cages and how many bees are ideally placed in these? Provide a reference that drones housed on their own, even with workers will survive. From COLloss and our experience, workers will only feed drones when there is a substantial number.

As stated at line 436, the bees were placed in wooden California queen cages. A reference showing that drones can survive this condition is not necessary, since our control data clearly show that drone survival over the observation period was 91-93% in the absence of an additional stressor (figure 1). This high survival may be because the bees were fed fondant, and the drones were observed eating the fondant. Drones struggle with syrup because of their lack of proboscis, but apparently, they are able to feed themselves solid food, which may have assisted their survival, despite the presence of workers.

24. 377 – what is the sample size of each treatment?

See point 21 above, with the new tables indicated.

25. 392 – what was the humidity and where were they kept? Optimal conditions are important for survival. While all bees were treated the same, adding a second stressor may increase mortality.

Unfortunately, we did not record the humidity. However, given that our experimental design included appropriate untreated controls and that the bees were monitored in the same environment, we do not think that this oversight invalidates the data. Indeed, being kept in a cage at all is itself an additional stressor, but the idea is that all the bees are experiencing the same cage, so the stressor tested is the variable condition.

26. 418 - please include the MS running parameters

We have now included the following details (line 482):

“As stated in McAfee et al.^{9,28}, the LC system included a fused-silica (5- μ m Aqua C18 particles (Phenomenex)) fritted 2-cm trap column connected to a 50-cm analytical column packed with ReproSil C18 (3- μ m C18 particles (Dr. Maisch)). The separation gradient ran from 5 to 35% Buffer B (80% acetonitrile, 0.1% formic acid) over 90 min, followed by a 15 min wash at 95% Buffer B (flow rate: 250 μ l/min). The instrument parameters were: scan from 150 to 2200 m/z, 100 μ s transient time, 10 μ s prepulse storage, 7 eV collision energy, 1500 Vpp Collision RF, a + 2 default charge state, 18 Hz spectral acquisition rate, 3.0 s cycle time and the intensity threshold was set to 250 counts.”

27. 419 – please include the proteome database, including date and search parameters used.

We already indicated that match between runs and LFQ options were enabled, but we have clarified that all other parameters remained on the default settings. The protein database is already included in the raw data archive available on MassIVE; however, we now provide a general description in the text as well (line 489):

“Mass spectrometry data was searched using MaxQuant (v 1.6.1.0) using default parameters, except that match between runs and label-free quantification were enabled. The data were searched against the honey bee reference proteome available on NCBI (based on the build HAv3.1, downloaded Nov 18th, 2019) with all honey bee virus and Nosema proteins included in the fasta file, which is available with the raw data hosted on MassIVE (www.massive.ucsd.edu; MSV000087818).”

28. 426 – the untreated bees were treated differently, in that they were not frozen prior to haemolymph extraction. Has effects of this variation to the protocol been tested? The variation may introduced large artifacts in protein abundance due to cell breakage during the freeze-thaw cycle. Please consider and discuss.

Thank you for bringing this up. The effect of frozen vs. fresh hemolymph collection has not been explicitly tested for proteomics, and we are unable to do so here due to other confounding factors (e.g., using a different instrument and LC system). However, this is essentially a validation test to investigate if the apparently high baseline levels of drone stress response proteins could be explained by the simple fact that drones may be more stressed than workers by being housed in a cage. As such, the data are meant to stand independently of the cage trial; we do not seek to compare fresh untreated drones to frozen acetone exposed drones, rather, we are comparing fresh drones to fresh workers to determine if the protein expression patterns corroborate what we observed comparing control drones to control workers in the cage trials. The data from the fresh, untreated drone/worker comparison largely agree with the data from the frozen control drone/worker comparison, supporting the notion that the strong baseline expression of putative stress response proteins is not an artifact of being caged. Since data from the two experiments are not being tested against one another, differences due to freezing are moot. We have clarified this with the following text (line 500):

“The purpose of this experiment is to determine if the differential expression observed between drones and workers in the cage trials could simply be an artifact of being caged. Here, fresh hemolymph from uncaged drones and workers was collected and compared

to determine if the same differential expression patterns are observed as between the control drones and workers in the prior cage trial.”

29. 427 – how many ng of protein were digested and ng of peptides analysed for the treatment groups?

Owing to different instrument sensitivities, 1 ug of peptides were analyzed by the QTOF whereas 200 ng of peptides were analyzed on the TIMS-TOF. We have now included this experimental detail.

30. 428 – what is the purpose of using a different version of MAXQuant for the treatments and the untreated? Were these samples not compared to the treatments? Why were not all samples run on the same version together?

We analyzed samples for the untreated drone/worker comparison some time after conducting the cage trial, when we realized that there was the possibility that the high levels of putative stress response proteins in drones could be an artifact of being caged. As such, the samples were run on a different instrument (the TIMS-TOF), which necessitated searching with a newer version of MaxQuant that included the option of LFQ for TIMS-TOF data (a relatively new feature). Since this is a stand-alone experiment and we can not conduct statistical tests between TIMS-TOF and QTOF data anyway, the data need not be searched all together. We have clarified the experimental purpose (see point 28 above) which we think makes this point more obvious.

31. 447 – how strong were the hives (number of brood frames, foundation) ? and how often were the patties replaced? Were the patties eaten in this period of time?

The colonies used in this experiment were overwintered colonies of similar size. As stated at line 528, the colonies were about the size of a ten-frame Langstroth deep hive body. More in-depth demographic measures were not performed. Patties were replaced daily (stated at Line 532). Significant portions of the patties were consumed (~50%) but patties were not entirely consumed during this period.

32. 455 – were the colonies from sister queens?

No, the queens in these colonies were not sister queens.

33. Line 465 – were these the same colonies that the adult fostering experiment took place in? If so can this please be made clearer?

Yes, these were the same colonies. We have added clarification to line 617.

34. 472 - 7-73 drones seems to be large range, please comment. We know that workers are more likely to feed drones when they are in large enough numbers in the cages, did the authors perform correlation analyses between drone number per cage and survival?

Having no prior knowledge that number of drones per cage was a factor in drone survival, and having direct experience to the contrary in prior experiments, this was unfortunately not something we considered during the conduct of the experiment. This range is high, but running a spearman's correlation on the number of drones per cage and the proportion survived shows a *negative* correlation (-0.38, $p=0.0484$) although one that we would reject with multiple comparisons correction; we therefore suggest in this experiment at least, the survival effect of the drones is not due to an imbalances in the cage stocking.

35. 482 – Please include components of Buffer D

The references provided describe the preparation in full, but we have recapitulated the components in brief in line 770.

“3.0 g/L D-glucose, 4.1 g/L KCL, 2.1 g/L NaHCO₃, and 24.3 g/L Na₃C₆H₅O₇.”

36. 503- 505 – was the original colony (mother of drones and workers) included in the statistical model? This needs to be clearer

We have added the following clarification (line 591):

“We did not have sufficient statistical power to test for colony source effects for the cage trial data; however, we included samples with roughly equal representation of each colony so as not to bias the analysis (see Table 3). For the uncaged worker and drone comparison, we were able to test for source effects between the two colonies sampled and none were identified, so this factor was not included in our statistical model.”

37. The Discussion seems feasible, although I would have liked to see a greater discussion of the different protein groups between workers and drones. Finally a greater discussion of the implications of neonicotinoid treatment on drones for colony strengths and future bee survival. Similar to the introduction many papers are not referenced and inclusion of these papers may allow a more interesting discussion.

To add to the discussion around proteins differentially expressed in workers and drones, we have included some details around adenylate kinase (line 333):

“One of the top five differentially expressed proteins was an adenylate kinase, an enzyme that plays a central role in cellular energy homeostasis, cell proliferation, and AMP-induced cell signalling; thus, it is a prime candidate to investigate as a master regulator of sex-specific metabolic rewiring.”

In light of the Bull *et al.* paper mentioned by the reviewer, we have added details around inducibility being poorly correlated with increased function (line 344):

“Conversely, these results also raise the question of how workers are so stress tolerant, in terms of survival, without launching an equally robust stress response. Indeed, we identified no differentially expressed proteins comparing workers treated with

imidacloprid to controls. While investigating worker responses to an entomopathogenic fungus, Bull et al. suggest that immune inducibility is not a reliable indicator of resistance, and it is possible that the same may be true with regard to abiotic stress responses. However, Bull et al. also propose that higher baseline expression of immune proteins in foragers relative to workers may explain why foragers are more resistant, despite exhibiting low inducibility. This creates a conundrum, because workers are generally more stress tolerant, and we identified both low inducibility and low baseline expression of stress response proteins relative to drones. On the whole, these data indicate that differential expression is difficult to interpret in the absence of protein activity, but the question remains: Why do drones express such high levels of putative stress response proteins, if they ostensibly remain inactive?"

On the point of including more discussion around the effect of neonicotinoids on drone quality in the context of honey bee health, we would prefer to avoid such discussion because the doses we tested are unrealistically high and, as stated in the text, are purely for the sake of investigating sex-biases in stress tolerance. We caution against extrapolating these data to learn about drone health in response to insecticide exposure, since this was not the purpose of the study and it was not designed in a way that would be informative for this goal. If we discuss the topic the reviewer is suggesting, it may appear somewhat disingenuous and we do not want to invite accusations of dosing pesticides at unreasonably high levels.

Reviewer #3 (Remarks to the Author):

This paper is well written (with some minor issues that I have pointed out below) and easy to comprehend. It studies a topic of interest to me: the relative ability of haploid drones and diploid worker honey bees to respond to and survive stresses. The data seem strong and the analyses appropriate. (My background is not in proteomics so I am not well qualified to comment on that aspect of the research.) I feel this paper adds considerably to our understanding of this topic. The 3 things that concern me the most that the authors should address, in my opinion, are:

We are thankful for these useful comments and for taking the time to review this manuscript in detail. Because of your thoughtful comments, we have reframed some of our arguments. We think the manuscript is improved with these additions and we appreciate the feedback.

1. The research is framed in the context of the "haploid susceptibility hypothesis". McAfee et al. in this paper framed that argument-- I believe incorrectly-- in terms of "deleterious recessive alleles". This hypothesis is described relatively briefly in the 2nd paragraph of the Introduction, and again at the beginning of the Discussion, but if a main goal of the experiments was to test this hypothesis, I think it should be addressed in somewhat more detail.

In light of comments also made by reviewer 1 and 2, we have substantially changed how we frame this hypothesis and the motivation behind this paper. Reviewer 1 had the impression that one of our main goals was to investigate drone immune responses, which we think is because that is the context in which the haploid susceptibility hypothesis was originally framed. And as reviewer 2 indicated, other existing papers on

drone sensitivity to stress were missing. In light of the fact that the haploid susceptibility hypothesis is discussed in the literature almost exclusively in the context of pathogens and parasites, and that there is a growing body of literature documenting susceptibility of haploid drones to abiotic stressors as well, we make the argument that this hypothesis should be expanded beyond the scope of parasites and pathogens. We also rephrase the hypothesis itself not as a consequence of deleterious recessive alleles, but from a lack of allelic diversity. We think, as the reviewer implies, that this is more true to how it was originally proposed.

Line 39:

“Existing research has shown that adult drone exposure of some pesticides³⁻⁸ and extreme temperatures⁹⁻¹² negatively impact drone fecundity. Other research has focused on drone exposure during larval development,¹³ but generally, little is known about how adult drone abiotic stress tolerance and their stress-mitigating responses compare to workers. Kairo et al. identified a negative effect of fipronil on drone fecundity but found no affect on survival; however, exact exposure levels were unknown because drones were exposed indirectly through foraging workers. Grassl et al.¹⁴ found that drones are more sensitive to thiamethoxam than workers, and our previous research shows that drones are more susceptible than workers to heat⁹.

This apparent biased sensitivity of male bees, which are haploid, to abiotic stressors may be in part explained by an extension of the haploid susceptibility hypothesis, which states that haploid individuals are more susceptible to pathogenic infections, since they have no opportunity for allelic diversity that comes with heterozygosity¹⁵. O'Donnell and Beshers, who proposed the haploid susceptibility hypothesis¹⁵, described the notion as a corollary to the well-known heterozygous advantage¹⁶. While existing examples of haploid susceptibility are described in the context of disease and parasites, in theory the notion should equally apply to abiotic stressors.”

...

“Given that drones are generally considered more sensitive to pesticide exposure than workers, which our data corroborates, we used these data to address our predictions that 1) drones have lower constitutive expression of relevant stress response proteins, and 2) that workers, but not drones, elevate detoxification enzymes in response to exposure. Our data suggest that drones have surprisingly strong baseline expression of putative stress response proteins, contrary to our expectations, causing us to re-evaluate exactly why drones, but not workers, are so intolerant to abiotic stress.

Regardless of the mechanism, we suggest that, in light of male sensitivity to abiotic stressors in honey bees, as documented here, the haploid susceptibility hypothesis may be expanded to include abiotic stress in addition to pathogens and parasites. Experiments investigating sex biases in other social insects are needed to determine how generalizable these observations are.”

2. A number of recent studies have demonstrated that pollen collected by honey bees almost always contains residues of pesticides. In a 2007 study by researchers at Penn State University wrote in the American Bee Journal (Frazier et al.), "In a total of 108 pollen samples analyzed, 46

different pesticides including six of their metabolites were identified." For this reason, I was surprised that there were no analyses of the pollen purchased to feed colonies or of the pollen stored in colonies rearing drones. I don't think this is serious for the study (all treatments had the same background level of contaminants in pollen), but this should be addressed.

This is a salient point. We did test control pollen patties from this supplier on 2 separate occasions (<https://doi.org/10.1016/j.chemosphere.2020.128183> and <https://doi.org/10.1038/s41598-020-80446-3>) and reported very low level detections of a limited number of contaminants. These detections did not pose meaningful risk and as you mentioned, would not have impacted this study. Due to resource limitations and timing these specific control patties were not tested, but we have no reason to think they would be different than previously observed.

3. The discussion seems quite speculative to me. This may be normal for proteomics research, where the functions of proteins are assumed based on other research. I would prefer to see the amount of speculation reduced.

We have conflicting feedback from reviewers on this point – Reviewer 2 asked for more discussion around potential roles of some of the differentially expressed proteins, which involves further speculation. We did add one sentence addressing a protein not previously discussed (adenylate kinase, line 334):

“One of the top five differentially expressed proteins was an adenylate kinase, an enzyme that plays a central role in cellular energy homeostasis, cell proliferation, and AMP-induced cell signalling; thus, it is a prime candidate to investigate as a master regulator of sex-specific metabolic rewiring.”

However, we refrained from further expansions of the discussion that would add speculation. We also removed some speculations (removed text: ***“Another explanation is that there may be underlying qualitative differences in drone stress proteins relative to workers. For example, despite finding an increased abundance of the glutathione-S-transferase in drones when compared to workers, drone glutathione-S-transferases may have a reduced detoxification activity towards pesticides. Qualitative differences in honey bee detoxification proteins have been previously reported and it has been found that enzyme abundance does not necessarily correlate with detoxification activity in honey bee workers³³. Similarly, large qualitative differences in another putative detoxification enzyme, esterase, have been identified in worker larvae from different breeding stocks while simultaneously finding no differences in esterase abundances²⁷.”***)

We hope that we have satisfied both reviewers with these changes.

Below are my more detailed comments. The first one adds to #1 above.

1. Lines 41-44: Second paragraph: Heterozygosity should confer additional benefits beyond just “heterozygous buffering of deleterious recessive alleles”. In fact, O’Donnell and Beshers stated in their paper on the haploid susceptibility hypothesis, “Genetic variation at the individual level may confer fitness advantages, particularly when codominant alleles at resistance loci contribute to the defence against pathogens. ... Any resistance trait that is affected by heterozygosity, including behavioural and

immunological responses, can contribute to haploid susceptibility". Heterozygous females could, for example, express two different genes with slightly different effects on heat stress, both beneficial when acting in concert. Haploid males would only express one of these alleles. I believe the authors have mis-interpreted the argument of O'Donnell and Beshers by focusing on "deleterious recessive alleles".

Thank you for pointing this out. Indeed, we did neglect to consider this additional aspect of the hypothesis, and we have reworded the text wherever it is mentioned. We rephrased the last sentence in the abstract to read (line 28):

"This suggests that drones' stress tolerance systems are fundamentally rewired relative to workers, and susceptibility to stress depends on more than simply gene dose or allelic diversity associated with hemi- and heterozygosity."

And in the introduction (line 47):

"This apparent biased sensitivity of male bees, which are haploid, to abiotic stressors may be in part explained by an extension of the haploid susceptibility hypothesis, which states that haploid individuals are more susceptible to pathogenic infections, since they have no opportunity for allelic diversity that comes with heterozygosity¹⁵. O'Donnell and Beshers, who proposed the haploid susceptibility hypothesis¹⁵, described the notion as a corollary to the well-known heterozygous advantage¹⁶. While existing examples of haploid susceptibility are described in the context of disease and parasites, in theory the notion could equally apply to abiotic stressors as well."

And in the discussion (line 303):

"The haploid susceptibility hypothesis states that haploid individuals are more susceptible to parasites and pathogens because their haploid state reduces allelic diversity while also providing no opportunity for heterozygous compensation should an unfavourable allele be possessed^{3,15}"

2. Line 102-112: The figure legend for Fig. 1 mentions 4 graphs (a-d), but the graph is only for 1a-c. It seems the first figure was deleted but the legend was never revised to reflect that change.

Thank you for pointing this out – you are right, we had previously structured this panel differently. We have updated the text to reflect the final version.

3. Line 154: For clarity, I suggest rewording "proportion dead in color" to "proportion dead presented in color".

Agreed and done.

4. Line 155: This currently reads "tests and size and fecundity which is somewhat confusing. I suggest deleting the first "and" adding a semicolon: "... with χ^2 tests; size and fecundity ...".

Agreed. We have edited the sentence to read (line 177):

"Mortality differences were evaluated with χ^2 tests, whereas size and fecundity were evaluated using linear mixed models"

4. Line 164: You wrote “for the other stressors”. Cold was a stressor in this study; but cannot be applied with “topical applications”. Please revise for clarity.

Agreed. The text now reads (line 187):

“Despite finding no clear effect of the pesticide cocktail treatments to whole colonies, sex-biased tolerance to imidacloprid was apparent. To investigate the molecular origin of workers’ and drones’ responses to pesticides, we performed...”

5. Line 170: For readers not very familiar with these types of studies, it would help to state what “FDR” stands for after its first usage.

Done.

6. Line 207: You have written: “... both sex and pesticide exposure comparisons.” Grammatically, I think there should be a hyphen between “pesticide-exposure” because both of those words together modify “comparisons. If so then, sex-exposure” should be hyphenated as well, resulting in “... both sex- and pesticide-exposure comparisons.”

Done, we have hyphenated “pesticide-exposure,” but kept “sex ... comparisons” the way it is because sex-exposure is not meaningful term here.

7. Line 234: First line in legend for Fig. 4. Can you add “to pesticides” for clarity? *“Differentially expressed proteins linked to stress responses to pesticides.*

Done.

8. Line 275: Delete the second comma, after stress. You have written: “drones are more susceptible to cold stress and pesticide stress than diploid worker bees, similar to heat stress, as has been demonstrated previously.” (With that comma present, you are saying that drones being more susceptible to cold and pesticide stress has been demonstrated previously! To be even clearer, I suggest you write: “similar to heat stress which has been demonstrated previously”.

Done.

9. Lines 318-320: You wrote: “Since we only quantified 654 protein groups, out of 1,452 identified proteins and still more proteins which exist below our limit of detection,” The comma should be deleted—it interferes with the interpretation of this clause. Also, the wording of the second part of this clause can be more clearly worded.

Agreed. It now reads (line 358):

“Since we only quantified 654 protein groups and many proteins exist below our limit of detection, it is possible that important stress response proteins were simply not quantified in our dataset”

10. Line 328: I was confused by the wording about heat shock proteins since you did not study heat shock in these experiments. I was waiting for you to cite your previous research—and then realized the bees had expressed heat shock proteins in response to pesticide treatments. This should be made clear at the start of this paragraph.

Agreed. It now reads (line 367):

“In the data presented here, we show that numerous heat-shock proteins were differentially expressed in drones relative to workers, as well as imidacloprid-exposed drones relative to unexposed drones. While drones expressed higher levels of HSP beta 1, HSP cognate 3....”

11. Line 334: When you write “temperature stress”, I think you are meaning heat stress and not cold stress?

Not necessarily. In other species, cold stress can actually stimulate expression of heat-shock proteins, too. However, there does not appear to be good data on this in honey bees, and the distinction is not important for the argument (the main message is that unexpected interactions between stressors could occur), so we have changed “temperature stress” to “heat stress.”

12. Line 345: Eastern honey bee: Have you mentioned this species previously? (I do not recall that you have.) I would suggest adding the scientific name, *Apis cerana*, here for readers who are not necessarily familiar with honey bees. (I acknowledge that you do mention the scientific name in the next sentence with the same reference #40, but I think the name should also occur where the common name of the species is given.)

Done.

13. Lines: 347-388: The word “data” is plural, to this should read “data show”.

Done.

14. Line 349: No comma is needed in “in Western honey bees, too.” Alternative, insert the word “also” after “enzyme” on line 348.

Done.

15. Line 377: Is a “wooden California queen cage” a “Benton cage”?

No, a California queen cage is a specific type of cage designed to contain a small number of bees (usually several workers and a queen, for shipping and ease of introduction into a colony). We tried to think of a more specific descriptor but they go by no other name and a quick search of “California queen cage” produces the correct item for any bee supply website, so we think the current description is sufficient.

16. Line 378: What is a “young, non-flying bee”? Were these bees collected from the brood nest of the hives, and presumed to be nurse bees because of their location in the hives?

That is correct. Also, because foragers can still occupy the brood nest and we were after nurses, we picked the bees up and dropped them ~12 inches to determine their propensity to fly. Those that did not fly after this agitation were considered “non-flying” bees and are highly likely to be nurses. We have now added this detail.

17. Line 381: What is “roughly equivalent representation from each colony”? Greater details are needed. Means and variances of numbers of bees in the tests, or max/min numbers, could be reported.

Agreed. We have now added a table showing how many bees were evaluated from each colony (Table 1):

Stressor	Sex	Colony	# Bees
Imidacloprid	Worker	Farm 002	26
Imidacloprid	Drone	Farm 002	30
Imidacloprid	Worker	Roof 002	20
Imidacloprid	Drone	Roof 002	13
Imidacloprid	Worker	Roof 004	19
Imidacloprid	Drone	Roof 004	13
Imidacloprid	Worker	SL NE	25
Imidacloprid	Drone	SL NE	32
Imidacloprid	Worker	SL SE	27
Imidacloprid	Drone	SL SE	29
Cocktail	Worker	Farm 002	26
Cocktail	Drone	Farm 002	31
Cocktail	Worker	SL NE	22
Cocktail	Drone	SL NE	26
Cocktail	Worker	SL SE	30
Cocktail	Drone	SL SE	26
Cold	Worker	Farm 002	17
Cold	Drone	Farm 002	18
Cold	Worker	SL NE	15
Cold	Drone	SL NE	16
Cold	Worker	SL SE	18
Cold	Drone	SL SE	13

19. Lines 282-399: I would recommend two paragraphs, one testing susceptibility to pesticides and a second one that explains cold stress experiments. The wording can be condensed in lines 393-395, by starting the new paragraph with “We also tested cold stress susceptibility by placing the cages with bees in a covered container in a 4 °C refrigerator for 0, 2, or 395 4 h and allowing all bees to recover as described above.”

We opted to divide the paragraph in a different way, since dividing it at the suggested sentence would lead to a single-sentence paragraph (since the cold-stress sentence can’t be the topic sentence for a paragraph that also describes recovery from all stressors). It now reads (line 452):

“...The imidacloprid solutions were produced by serial dilution of the technical chemical acquired from Chem Service Inc. (West Chester, PA). We tested doses of 0, 1, 10, and 100 ppm. In addition to the pesticide challenge, we also tested acute cold stress susceptibility by placing the caged bees in a covered container in a 4 °C refrigerator for 0, 2, or 4 h.

After all treatments, bees were allowed to recover for two days at room temperature in the dark, and were provided with two drops of water (~100 µl) per day. After the two day

stress recovery period, we counted the number of bees that were alive and dead. Workers and drones from the highest sublethal doses tested (10x cocktail, and 10 ppm imidacloprid) were euthanized ..."

20. Line 409: What is "4 volumes of ice-cold acetone"? The volumes added could be 4 X 10 microliters, or variable volumes, etc. Perhaps this is standard terminology, but it is not clear to me.

We have clarified that the final acetone concentration was 80% (adding 4 volumes, or 4x the sample volume, achieves this, line 472):

"Clarified solution was precipitated using acetone (final acetone concentration = 80%, incubated at -20 °C overnight)"

21. Line 418: these words seem misplaced: "in randomized order". I would suggest "and analyzed in randomized order on..."

Done.

22. Line 419: "Data" is plural: "data were searched."

Done.

23. Line 434: I would suggest changing "based off of" to "based on"

Done.

24. Line 440: Was the pollen purchased from Glory Bee Foods tested for pesticides? If not, this seems to be a serious oversight.

It was tested, and determined to have very low/trace levels of contaminants in our prior experiments. These specific pollen patties were not tested but we have no reason to expect the baseline levels to be different from those we reported previously (ref 26 and 27). We now refer to these references in the methods.

25. Lines 451-452: At what intervals were the pollen patties replaced to maintain continual exposure with quality pollen? Perhaps the duration of feeding was too brief to require replacement of the pollen patties, but the duration of feeding is not stated.

Pollen patties were replaced daily for each colony (stated at line 569). Pollen patty feeding commenced 28d before and during drone development within drone larval rearing colonies. This information was stated at line 577 and line 584. This time frame ensures that an entire brood cycle of workers was exposed to our treatment during development before starting drone rearing.

26. Line 454: I suggest changing "drones all sampled" to "all drones sampled" (unless I have not understood the wording here). How does this journal prefer reference to dates? I am in the habit of writing "3-7 May".

We implemented the reviewer's suggestions on line 572.

27. Line 455: I would suggest rewording “two colony sources” to “two source colonies” (the drones were collected from colonies).

We implemented the reviewer’s suggestions on line 573.

28. Line 457: My apology if I missed this detail, but what is a “rearing cage”?

We have corrected this typo in the manuscript on line 575.

29. Line 466: You state that colonies were fed pollen “for at least 28 days”. This suggests that the duration of exposure to the pollen varied between colonies. If so, please provide better details.

The minimum exposure was 28 days prior to development but there were slight differences depending on the timing of egg laying by each respective queen. The latest a queen layed was 32 days after starting pollen feeding which is still within an entire worker brood cycle. This additional detail was added to line 584.

30. Line 467: There are published details of the timing of eye colour changes during development of workers and drones. What colour states (stages of drone development) were the drone eyes when you removed the frames of drone brood?

There are such references. We have added details to our methodology (lines 724-725). In short, we collected drone frames when the pupal eyes had reached a dk purple/black coloration (or about 2 d from emergence).

31. Line 468-469: “Daily bees were removed” should be changed to “Bees were removed daily”.

Done.

32. Line 470: Bees were placed in an “untreated colony”. Where was it located? How do we know that bees in it had not brought pesticides used in the surrounding area into the colony? Was pollen from it tested? I understand that all bees placed into it experienced the same conditions, but how free was it of pesticides?

The source colonies used to rear drones were either fed treated or untreated pollen and colonies in both groups were limited by a pollen excluder to help encourage consumption of our experimental pollen patties (as described in the methods section). Alternatively, the foster colony used in experiment two (testing the impacts of exposure during drone development) was untreated and did not receive any pesticide treated pollen, or any additionally pollen at all. This foster colony did not have a pollen excluder trap either. Additional clarification was added to this section at line 623.

Bees forage for miles around a colony and it would be very difficult to restrict this behavior without introducing additional confounding variables. As a result, the presence of some chemicals within the resources being collected by the foster colony is inevitable. Ultimately, it is the dose that makes the poison and based on previous in-hive samples from colonies at this bee yard we feel that there is not a significant amount of risk from pesticides the foster-colony may have collected from the ambient surrounding landscape.

33. Line 476: “head and thoraxes”: singular and plural. I would suggest “head and thorax” (singular, to match “head and abdomen” in line 478).

Done.

34. Line 477 does not read well. Needs editing.

Agreed. It now reads (line 561):

“The head and thorax were photographed, then drones were dissected to obtain their mucus glands. The seminal vesicles were cut free from the testicles and ejaculatory duct, then photographed. Finally, the head, wings, and abdomen were cut free from the thorax and legs and these were weighed.”

35. Drones can be large or small (reared in drone vs. worker cells). There is no mention of rejection or nonexistence of small drones.

Indeed and size is well-known to covary with reproductive output. However, the actual occurrence of worker-cell reared drones is small in queenright colonies, which these colonies were. Additionally, the drones used in these studies were obtained from drone comb as stated in lines 701 and 719. There is other work by the authors and several other lab groups that directly addresses cell size. This study rather focuses on more acute environmental phenomena. Our measurements of size in this study are therefore limited to controlling for size effect on reproduction with our statistical analyses.

36. Line 496: This construction is complicated! Please simplify: “Adult drone topical exposure and cold exposure survival counts...”

Done. This now reads (line 583):

“Adult drone topical- and cold-exposure survival counts were evaluated by logistic regression in R (v 3.6.0)⁵²”

37. Line 504: I suggest this should be hyphenated: “two-group”.

Done.

38. Line 552-553: Code or codes or coding? Please revise so subject and verb agree in number (“any code is available” vs. “any coding is available” vs. “all codings are available”, etc.).

Done.

39. Line 537: NSERC “of Canada” (add these words for clarity).

Done.

40. References: Scientific names should be italicized. The very first reference has *Apis mellifera* without italics. Same issue with references #7, #9, #12, #13, and many others. In fact, it seems that no scientific names are italicized. I checked a paper published in 2020 and the scientific name was italicized.

There should be consistency in formatting (e.g. abbreviations of journal names as in reference #8 and #22; capitalization of title as in #19; capitalization of journal name as in #18 and #20). Please review all references and make sure they are 100% accurate. This is not a task for reviewers.

Apologies, either this lack of italics was a glitch with the reference managing software, or it is part of the Nature citation style rules within Endnote. We have corrected the species names manually. The journal abbreviations are formatted according to the Nature endnote style, which follows specific rules. Should this paper be accepted for publication we will look deeper into this problem but if not, we may need to reformat all the references anyway, so we wish to defer this task until acceptance is known.

41. Supplementary File 1

I opened the pdf file first. There are issues with it. It would help if there was some information about what the three data files are. There are 3 sets of info that run together. The first line with the headings for the 2nd set of data run together, making it nearly impossible to decipher. In fact, in converting from the excel file to a pdf file, the headings seem to have been cut off. On pages 29-32 there are spaces in the data that seem like they should be removed.

Then I opened the "source file". It is VERY different from the pdf file, with much more information. I question the value of the pdf file. (Perhaps I did not correctly review these files.)

We cannot explain this issue, since we upload only the data tables in excel format, and the built-in submission pdf-builder does the rest (there are no options the user can select to either convert or not convert). This issue appears to be one with the submission platform, not anything within our control. The Excel file is the one that will be linked to the final version of the manuscript online.

42. Related to this, in the Reporting Summary, Under "Data", the legends for all the supplementary data files have been repeated, word for word. Table S1-6 are summarized, then summarizes again EXCEPT the two descriptions for Table S-1 differ.

Thank you for catching this error! We have deleted the redundant sections.

REVIEWERS' COMMENTS:

Reviewer #1 (Remarks to the Author):

In general, the authors satisfactorily answered the questions raised. The only unsolved problem was about collecting hemolymph from frozen and dead insects. This technical problem should be considered as a limitation, recognizing the implication of this for data interpretation or the collection of proteins detected throughout the work.

Reviewer #2 (Remarks to the Author):

the authors have made great improvements to the manuscript and it is now a clear and as before very interesting read. contributing new information to the field of drone susceptibility.

Reviewer #3 (Remarks to the Author):

When I reviewed this manuscript the first time I was positive about the research and the writing. I will not repeat the comments I made about it.

In the revised manuscript, the authors have done a good job of fixing most of the issues identified by the 3 reviewers. The manuscript now reads much better and in my opinion is a valuable contribution about drones and their responses to stressors. They should be commended for their efforts to address the comments of the reviewers.

I have made comments and changes directly on the manuscript itself. It may seem that I have made a lot of corrections that I should have made on the earlier draft. I apologize for that. My explanation is that until a manuscript is in really good shape structurally and grammatically, I have difficulty detecting minor issues.

My main comments are summarized here:

- 1) Intra-colony relatedness values given on lines 72 & 73 are not correct. Drones are related to their sisters by 1/4 (on average) and to their mother by 1/2 (i.e., being haploid, they only inherit one set of chromosomes of the queen's two sets of chromosomes).
- 2) Figure 2: Avoid red and green objects in figures. 8% of human males of European descent are red-green color blind. This may be an issue in Figure 4 & 5 as well, where the two colors are close to red and green.
- 3) Two reviewers questioned what California queen cages are. Please give the dimensions and perhaps a website that depicts such a cage rather than saying that it is easy to look this info up on the internet. While this may be a commonly used type of cage in N. America, I'm not sure if the same is true for Europe, Asia, Africa, S. America, and Australia.
- 4) Table 2. There seems to be a subscript "b" after "2,4-DMPF" but no explanation in the table legend.
- 5) Line 540: The authors explained that the pollen purchased from Glorybee Foods had been analyzed for pesticide residues previously. That information has not been presented in the methods section.
- 6) Lines 552-553: The wording states that mass lost from pollen patties is due to consumption of the pollen. Do the bees never store the pollen-patty materials in their combs? (i.e., is all material removed from the pollen patty consumed immediately?)
- 7) There are still numerous minor grammatical errors, as well as some inconsistency in the format of references. (Did all 5 authors really review this manuscript critically and approve it, as has been indicated?) I have made corrections or suggested alternative wording where I detected issues.

Reviewer #3 (Remarks to the Author):

When I reviewed this manuscript the first time I was positive about the research and the writing. I will not repeat the comments I made about it.

In the revised manuscript, the authors have done a good job of fixing most of the issues identified by the 3 reviewers. The manuscript now reads much better and in my opinion is a valuable contribution about drones and their responses to stressors. They should be commended for their efforts to address the comments of the reviewers.

I have made comments and changes directly on the manuscript itself. It may seem that I have made a lot of corrections that I should have made on the earlier draft. I apologize for that. My explanation is that until a manuscript is in really good shape structurally and grammatically, I have difficulty detecting minor issues.

Thank you! We did not receive a marked up manuscript file, but we have conducted a thorough line edit on the present document and caught numerous inconsistencies and errors.

My main comments are summarized here:

1) Intra-colony relatedness values given on lines 72 & 73 are not correct. Drones are related to their sisters by 1/4 (on average) and to their mother by 1/2 (i.e., being haploid, they only inherit one set of chromosomes of the queen's two sets of chromosomes).

Apologies for this oversight. The relatedness value for the drone to queen is actually correct (since 100% of the drone's genes are in the queen, he is 100% related to her, but she is only 50% related to him). The drone-worker relatedness was incorrect, as you say. We have reworded the sentence as follows:

"They also experience indirect selection through their sister workers (on average, 25% of a drone's genes are present in his sisters) and mother queen (100% of a drone's genes are present in his mother)."

2) Figure 2: Avoid red and green objects in figures. 8% of human males of European descent are red-green color blind. This may be an issue in Figure 4 & 5 as well, where the two colors are close to red and green.

We have changed the palette in all three figures to a color-blind friendly version.

3) Two reviewers questioned what California queen cages are. Please give the dimensions and perhaps a website that depicts such a cage rather than saying that it is easy to look this info up on the internet. While this may be a commonly used type of cage in N. America, I'm not sure if the same is true for Europe, Asia, Africa, S. America, and Australia.

We have revised the text:

"On Day 5, they were collected and placed in wooden California mini queen cages (approximately 7 cm x 2.5 cm x 1.3 cm with one open face covered by a mesh screen) containing fondant in a candy tube."

4) Table 2. There seems to be a subscript "b" after "2,4-DMPF" but no explanation in the table legend.

We have added:

“2,4-Dimethylphenyl formamide (Amitraz degradate)”

5) Line 540: The authors explained that the pollen purchased from Glorybee Foods had been analyzed for pesticide residues previously. That information has not been presented in the methods section.

We have updated the text with this information:

“We have obtained pollen from this supplier (Glorybee Foods Inc.) previously and background pesticide residue testing showed very low detections of a limited number of contaminants, which did not pose meaningful risk to the bees (see Milone et al. (2021)²⁹ and Milone and Tarpy (2021)²⁷, and methods therein).”

6) Lines 552-553: The wording states that mass lost from pollen patties is due to consumption of the pollen. Do the bees never store the pollen-patty materials in their combs? (i.e., is all material removed from the pollen patty consumed immediately?)

Although to our knowledge, no one has explicitly tested this, anecdotal evidence of this abounds amongst beekeepers (e.g. <https://www.beeeculture.com/andy-oliver-honeybee-nutrition-part-4/> - see “Using Fluorescent Tracer To Track Protein Sub Distribution In The Hive”). Nevertheless, the statement that bees consume the pollen directly is a presumption, and we have stated it as such at the beginning of the relevant methods section:

“Honey bees do not normally store pollen patty supplement and loss of pollen patty mass is presumed to represent consumption.”

7) There are still numerous minor grammatical errors, as well as some inconsistency in the format of references. (Did all 5 authors really review this manuscript critically and approve it, as has been indicated?) I have made corrections or suggested alternative wording where I detected issues.

We have reviewed all references in detail and made numerous formatting corrections.

We did not receive an attachment with this review but we have now thoroughly revised the article for grammatical errors as suggested.